# Cardinality constrained submodular maximization for random streams

**Paul Liu**
Department of Computer Science
Stanford University
paul.liu@stanford.edu

**Aviad Rubinstein**
Department of Computer Science
Stanford University
aviad@stanford.edu

**Jan Vondrák**
Department of Mathematics
Stanford University
jvondrak@stanford.edu

**Junyao Zhao**
Department of Computer Science
Stanford University
junyaoz@stanford.edu

## Abstract

We consider the problem of maximizing submodular functions in single-pass streaming and secretaries-with-shortlists models, both with random arrival order. For cardinality constrained monotone functions, Agrawal, Shadravan, and Stein [ASS19] gave a single-pass $(1 - 1/e - \varepsilon)$-approximation algorithm using only linear memory, but their exponential dependence on $\varepsilon$ makes it impractical even for $\varepsilon = 0.1$. We simplify both the algorithm and the analysis, obtaining an exponential improvement in the $\varepsilon$-dependence (in particular, $O(k/\varepsilon)$ memory). Extending these techniques, we also give a simple $(1/e - \varepsilon)$-approximation for non-monotone functions in $O(k/\varepsilon)$ memory. For the monotone case, we also give a corresponding unconditional hardness barrier of $1 - 1/e + \varepsilon$ for single-pass algorithms in randomly ordered streams, even assuming unlimited computation.

Finally, we show that the algorithms are simple to implement and work well on real world datasets.

## 1 Introduction

Over the past few decades, submodularity has become recognized as a useful property occurring in a wide variety of discrete optimization problems. Submodular functions model the property of diminishing returns, whereby the gain in a utility function decreases as the set of items considered increases. This property occurs naturally in machine learning, information retrieval, and influence maximization, to name a few (see [IKBA20] and the references within).

In many settings, the data is not available in a random-access model; either for external reasons (customers arriving online) or because of the massive amount of data. When the data becomes too big to store in memory, we look for *streaming algorithms* that pass (once) through the data, and efficiently decide for each element whether to discard it or keep it in a small buffer in memory.

In this work, we consider algorithms that process elements arriving in a random order. Note that the classical greedy algorithm which iteratively adds the best element [NWF78] cannot be used here, and hence we must look for new algorithmic techniques. The motivation for considering random element arrival comes from the prevalence of submodularity in big-data applications, in which data is often logged in batches that can be modelled as random samples from an underlying

35th Conference on Neural Information Processing Systems (NeurIPS 2021).

distribution.[1] The problem of streaming submodular maximization has recently received significant attention both for random arrival order and the more pessimistic worst-case order arrival [ASS19, AEF+20, AF19, BMKK14, BV14, CGQ15, FKK18, FNSZ20, HKMY20, IV19, KMZ+19, MV19, NTM+18, HTW20, Sha20].

## 1.1 Submodular functions and the streaming model

Let $f : 2^E \to \mathbb{R}_+$ be a non-negative set function satisfying $f(S \cup \{e\}) - f(S) \geq f(T \cup \{e\}) - f(T)$ for all $S \subseteq T \subseteq E \setminus \{e\}$. Such a function is called *submodular*. For simplicity, we assume $f(\emptyset) = 0$.[2] We use the shorthand $f(e|S) := f(S \cup \{e\}) - f(S)$ to denote the *marginal* of $e$ on top of $S$. When $f(S) \leq f(T)$ for all $S \subseteq T$, $f$ is called *monotone*. We consider the following optimization problem:

$$OPT := \max_{|S| \leq k} f(S)$$

where $k$ is a cardinality constraint for the solution size.

Our focus on submodular maximization in the streaming setting. In this setting, an algorithm is given a single pass over a dataset in a streaming fashion, where the stream is a some permutation of the input dataset and each element is seen once. The stream is in *random order* when the permutation is uniformly random. When there are no constraints on the stream order, we call the stream *adversarial*.

At each step of the stream, our algorithm is allowed to maintain a buffer of input elements. When a new element is streamed in, the algorithm can choose to add the element to its buffer. To be as general as possible, we assume an oracle model — that is, we assume there is an oracle that returns the value of $f(S)$ for any set $S$. The decision of the algorithm to add an element is based only on queries to the oracle on subsets of the buffer elements. The algorithm may also choose to throw away buffered elements at any given time. The goal in the streaming model is to minimize memory use, and the complexity of the algorithm is the maximum number of input elements the algorithm stores in the buffer at any given time.

For the oracle model, an important distinction is between *weak oracle* access or and *strong oracle* access. In the *weak oracle* setting, the algorithm is only allowed to query sets of feasible elements (sets that have cardinality less than $k$). In the *strong oracle* setting however, the algorithm is allowed to query any set of elements. All our results apply to both the weak and strong oracle models.

Our aim will be to develop algorithms that only make one pass over the data stream, using $\tilde{O}_\varepsilon(k)$ memory, where $k$ is the maximum size of a solution set in $\mathcal{I}$. We assume that $k$ is small relative to $n := |E|$, the size of the ground set.

## 1.2 Our contributions

On the algorithmic side, we give $(1 - 1/e - \varepsilon)$ and $(1/e - \varepsilon)$ approximation for cardinality constrained monotone and non-monotone submodular maximization respectively, both using $O(k/\varepsilon)$ memory. The monotone result has an exponential improvement in memory requirements compared to Agrawal et al. [ASS19] (in terms of the dependence on $\varepsilon$), while the non-monotone result is the first to appear in the random-order streaming model, and improves upon the best known polynomial-time approximations under adversarial orders [AEF+20]. The algorithms are extremely simple to implement, and perform well on real world data (see Section 5), even compared to *offline* greedy algorithms.

On the hardness side, we prove that a $(1 - 1/e + \varepsilon)$-approximation for monotone submodular maximization would require $\Omega(n)$ memory (even with unlimited queries and computational power). This improves the hardness bound of $7/8$ from [ASS19].

## 1.3 Related work

Prior work on this problem has focused on both the adversarial and random-order streaming setting. Algorithmic and hardness results further depend on whether the function $f$ is monotone or non-

---

[1] Note that the random order assumption generalizes the typical assumption of i.i.d. sampling stream elements from a known distribution, as the random order assumption applies equally well to unknown distributions.

[2] Submodular functions $f$ with $f(\emptyset) > 0$ only improves the approximation ratio of our algorithms.

|  | adversarial order | random order |
|---|---|---|
| monotone | $\geq 1/2 - \varepsilon$ ($O(k/\varepsilon)$ [KMZ$^+$19]) 
 $\leq 1/2 + \varepsilon$ (W+S [NTM$^+$18, FNSZ20]) 
 $\leq 2/(2 + \sqrt{2})$ (S [HKMY20]) | $\geq 1/2 + c, c < 10^{-13}$ ($O(k \log k)$ [NTM$^+$18]) 
 $\geq 1 - 1/e - \varepsilon$ ($O(k \exp(\mathrm{poly}(1/\varepsilon)))$ [ASS19]) 
 $\leq 1 - 1/e + \varepsilon$ (S) 
 $\geq 1 - 1/e - \varepsilon$ ($O(k/\varepsilon)$, this paper) |
| non-mono. | $\geq 1/2 - \varepsilon$ (unbounded comp, $O(k/\varepsilon^2)$ [AEF$^+$20]) 
 $\geq 0.2779$ ($O(k)$ [AEF$^+$20]) 
 $\geq 0.1715$ ($O(k)$ [FKK18]) 
 $< 1/2 + \varepsilon$ (W [AEF$^+$20]) | $\geq 1/e$ ($O(k/\varepsilon)$, this paper) |

Table 1: A survey of results on monotone submodular streaming. Our contributions are highlighted in magenta. W = weak oracle, S = strong oracle, complexities refer to memory use. Lower-bounds propagate downwards in the table. Note that our polynomial-time algorithms require only weak oracle access, while our hardness results hold even for strong oracle.

monotone, and whether $f$ has explicit structure (e.g. such as by presenting a set system for $f$ in the coverage case), or accessible only via oracle queries. Table 1 describes all the relevant results.

**Algorithmic results.** Submodular maximization in the streaming setting was first considered by Badanidiyuru et al. [BMKK14] who gave a $(1/2 - \epsilon)$-approximation in $O(k \log k/\epsilon)$ memory for monotone submodular functions under a cardinality constraint, using a thresholding idea with parallel enumeration of possible thresholds. This work led to a number of subsequent developments with the current best being a $(1/2 - \epsilon)$-approximation in $O(k/\epsilon)$ memory [KMZ$^+$19]. It turns out that the factor of $1/2$ is the best possible in the adversarial setting (with a weak oracle), but an improvement is possible in the random order model (the input is ordered by a uniformly random permutation). This was first shown by Norouzi-Fard et al. [NTM$^+$18], who proved that the $1/2$-hardness barrier for cardinality constraints can be broken, exhibiting a $(1/2 + c)$-approximation in $O(k/\varepsilon \log k)$ memory where $c < 10^{-13}$. In a breakthrough work, Agrawal, Shadravan, and Stein [ASS19] gave a $(1 - 1/e - \varepsilon)$-approximation using $k \, 2^{\mathrm{poly}(1/\varepsilon)}$ memory and running time. We note that this is arbitrarily close to the optimal factor of $1 - 1/e$, but the algorithm is not practical, due to its dependence on $\varepsilon$ (even for $\epsilon = 0.1$, the resulting constants are astronomical).

**Lower bounds.** A few lower bounds are known for monotone functions in the *adversarial order model*: with a weak oracle, any $(1/2 + \epsilon)$-approximation would require $\Omega(n/k)$ memory [NTM$^+$18]. Under a strong oracle, a lower bound of $\Omega(n/k^3)$ memory for any $(1/2 + \epsilon)$-approximation algorithm was shown in a recent paper by Feldman et al. [FNSZ20]. Another recent lower bound was proved by McGregor and Vu [MV19]: a $(1 - 1/e + \varepsilon)$-approximation for coverage functions requires $\Omega(n/k^2)$ memory (this lower bound holds for explicitly given inputs, via communication complexity; we note that this is incomparable to the computational $(1 - 1/e + \varepsilon)$-hardness of maximum coverage [Fei98]). For non-monotone functions, Alaluf et al. [AEF$^+$20] proved an $\Omega(n)$ memory lower bound for the adversarial order model with unbounded computation.

In the random-order model, Agrawal et al. [ASS19] show that beating $7/8$ (for monotone submodular functions) requires $\Omega(n)$ memory. In contrast, we show that same construction as McGregor and Vu [MV19] also applies to randomly ordered streams: $(1 - 1/e + \varepsilon)$ for coverage functions requires $\Omega(n/k^2)$ memory even in the random-order model.

**Submodular maximization in related models.** A closely related model is the secretary with shortlists model [ASS19], where an algorithm is allowed to store a shortlist of more than $k$ items (where $k$ is the cardinality constraint). Unlike the streaming model however, once an element goes into the shortlist, it cannot be removed. Then, after seeing the entire stream, the algorithm chooses a subset of size $k$ from the shortlist and returns that to the user. We note that the algorithms developed in this paper apply almost without modification to the shortlists model.

### 1.4 Overview of our techniques

**Main algorithmic techniques.** The primary impetus for our algorithmic work was an effort to avoid the extensive enumeration involved in the algorithm of Agrawal et al. [ASS19] which leads to memory requirements *exponential* in poly$(1/\varepsilon)$.

To make things concrete, let us consider the input divided into disjoint *windows* of consecutive elements. The windows containing actual optimal elements play a special role — let's call them *active windows* — these are the windows where we make quantifiable progress. When the stream is randomly ordered, we would ideally like to have each new element sampled independently and uniformly from the input. This leads to the intuition that the optimal elements are evenly spread out through all the windows. This cannot be literally true, since conditioned on the history of the stream, some elements have already appeared and cannot appear again. However, a key idea of Agrawal et al. [ASS19] allows us to circumvent this by reinserting the elements that we have already seen and that played a role in the selection process. What needs to be proved is that elements that were not selected can still appear in the future, conditioned on the history of the algorithm; that turns out to be true, provided that our algorithm operates in a certain greedy-like manner.

To ensure progress was made regardless of the positioning of the optimal elements, previous work made use of exponentially large enumerations to essentially guess which windows the optimal elements arrive in. Where we depart from previous work is the way we build our solution. The idea is to use an evolving family of solutions which are updated in parallel, so that we obtain a quantifiable gain regardless of where the optimal elements arrived. Specifically, we grow $k$ solutions in parallel, where solution $L_i$ has cardinality $i$. In each window, we attempt to extend a collection of solutions $L_i$ (for varying $i$) by a new element $e$; if $e$ is beneficial on average to every $L_i$ in the collection, we replace each $L_{i+1}$ with the new solution $L_i \cup \{e\}$. Regardless of which windows happen to be active, we will show that the average gain over our evolving collection of solutions is analogous to the greedy algorithm. This is the basis of the analysis that leads to a factor of $1 - 1/e$.

In addition to our candidate solutions $L_i$, we maintain a pool of elements $H$ that our algorithm has ever included in some candidate solution. We then use $H$ to reintroduce elements artificially back into the input; this makes it possible to assume that every input element still appears in a future window with the same probability, which is key to the probabilistic analysis leading to $1 - 1/e$.

**Non-monotone functions.** Our algorithm for non-monotone submodular functions is similar, with the caveat that here we also have to be careful about not including any element in the solution with large probability. This is an important aspect of the randomized greedy algorithm for (offline) non-monotone submodular maximization [BFNS14] which randomly includes in each step one of the top $k$ elements in terms of marginal values. We achieve a similar property by choosing the top element from the current window and a random subset of the pool of prior elements $H$.

**Hardness results.** Our hardness instances have the following general structure: there is a special subset of $k$ *good* elements, and the remaining $n - k$ elements are *bad*. The good elements are indistinguishable from each other, and ditto for the bad elements. In the monotone case, any $k$ bad elements are a $(1 - 1/e)$-factor worse than the optimal solution ($k$ good elements). Suppose furthermore that for parameter $r > 0$, as long as we never query the function on a subset with $\geq r$ good elements, the good elements are indistinguishable from bad elements. The only way to collect $\geq r$ good elements in the memory buffer is by chance – until we've collected the required number of good elements, they are indistinguishable from bad elements, so the subset in the memory buffer is random. The classic work of Nemhauser and Wolsey [NW78] constructs a pathological monotone submodular function with $r = \Omega(k)$, which we use to prove that without memory $\Omega(n)$ the algorithm cannot beat $1 - 1/e$. McGregor and Vu [MV19] construct a simple example of a coverage function with $r = 2$, which we use for our $\Omega(n/k^2)$ bound. For an exponential-size ground set, we extend their construction to $r = 3$ which translates to the improved lower bound of $\Omega(n/k^{3/2})$.

## 2 A $(1 - 1/e - \varepsilon)$-approximation in $O(k/\varepsilon)$ memory

In this section, we develop a simple algorithm for optimizing a monotone submodular function with respect to a cardinality constraint. Due to space limitations, we focus on the intuition behind the results. Full proofs can be found in the supplementary.

Our algorithm begins by randomly partitioning the stream $E$ into $\alpha k$ contiguous *windows* of expected size $n/(\alpha k)$, where $\alpha > 1$ is a parameter controlling the memory dependence and approximation ratio. This is done by generating a random partition according to Algorithm 1. As the algorithm progresses, it maintains $k$ partial solutions, the $\ell$-th of which contains exactly $\ell$ elements. Within each window we process all the elements independently, and choose one candidate element $e$ to extend the partial solutions by. We then add $e$ to a collection of partial solutions $L_\ell$ at the end of the window. The range of partial solution sizes $\ell$ that we use roughly tracks the number of optimal elements we are expected to have seen so far in the stream.

Intuitively, our algorithm is guaranteed to make progress on windows that contain an element from the optimal solution $O$. Let us loosely call such windows *active* (a precise definition will be given later). Of course, the algorithm never knows which windows are active. However, the key idea of our analysis is that we are able to track the progress that our algorithm makes on active windows. Since the input stream is uniformly random, intuitively we expect to see $i/\alpha$ optimal elements after processing $i$ windows. With high probability, the true number of optimal elements seen will be in the range $\mathcal{R} := [i/\alpha - \tilde{O}(\sqrt{k}), i/\alpha + \tilde{O}(\sqrt{k})]$. By focusing on the average improvement over levels in $\mathcal{R}$, we can show that each level $\ell$ in this range gains $\frac{1}{k}(f(O) - f(L_{\ell-1}))$ in expectation, whenever a (random) optimal element arrives.

For the analysis to work, ideally we would like each arriving optimal element to be selected uniformly among all optimal elements. This is not true conditioned on the history of decisions made by the algorithm. However, we can remedy this by re-inserting elements that we have selected before and subsampling the elements in the current window, with certain probabilities. A key lemma (Lemma 2.2) shows why this works, since the elements we have never included might still appear, given the history of the algorithm. Our basic algorithm is described in Algorithm 2, with the window partitioning procedure described in Algorithm 1.

---

**Algorithm 1** Partitioning stream $E$ into $m$ windows.

1: Draw $|E|$ integers uniformly from $1, \ldots, m$
2: $n_i \leftarrow$ # of integers equal to $i$
3: $t_i \leftarrow \sum_{1 \le j < i} n_i$
4: **for** $i = 1, \ldots, m$ **do**
5: $\quad w_i \leftarrow$ elements $t_i$ to $t_i + n_i$ in $E$
6: **return** $\{w_1, w_2, \ldots, w_m\}$

---

**Algorithm 2** MONOTONESTREAM$(f, E, k, \alpha)$

1: Partition $E$ into windows $w_i$ for $i = 1, \ldots, \alpha k$ with Algorithm 1.
2: $L_\ell^0 \leftarrow \emptyset$ for $\ell = 0, 1, \ldots, k$
3: $H \leftarrow \emptyset$
4: **for** $i = 1, \ldots, \alpha k$ **do**
5: $\quad C_i \leftarrow w_i$
6: $\quad L_\ell^i \leftarrow L_\ell^{i-1}$ for $\ell = 0, 1, \ldots, k$
7: $\quad$ Sample each $e \in H$ with probability $\frac{1}{\alpha k}$ and add to a set $R_i$
8: $\quad C_i \leftarrow C_i \cup R_i$
9: $\quad z_l, z_h \leftarrow \max\{0, \lfloor i/\alpha \rfloor - 20\alpha\sqrt{k \log k}\}, \min\{k, \lceil i/\alpha \rceil + 20\alpha\sqrt{k \log k}\}$
10: $\quad e^\star \leftarrow \operatorname{argmax}_{e \in C_i} \sum_{\ell=z_l}^{z_h} f(e|L_\ell^{i-1})$
11: $\quad$ **if** $\sum_{\ell=z_l}^{z_h} f(L_\ell^{i-1} \cup \{e^\star\}) > \sum_{\ell=z_l}^{z_h} f(L_{\ell+1}^{i-1})$ **then**
12: $\quad\quad H = H \cup \{e^\star\}$
13: $\quad\quad L_{\ell+1}^i \leftarrow L_\ell^{i-1} \cup \{e^\star\}$ for all $\ell \in [z_l, z_h]$
14: $\quad\quad$ **for** $\ell = 1, 2 \ldots, k$ **do**
15: $\quad\quad\quad$ **if** $f(L_\ell^i) \ge f(L_{\ell+1}^i)$ **then**
16: $\quad\quad\quad\quad L_{\ell+1}^i \leftarrow L_\ell^i + \operatorname{arg max}_{e \in L_{\ell+1}^i} f(e|L_\ell)$
17: **return** $\operatorname{arg max}_{1 \le \ell \le k} f(L_\ell^{\alpha k})$

---

In its most basic implementation, Algorithm 2 requires $O(\alpha k + k^2)$ memory (to store $H$ and $L_\ell^i$'s for $\ell = 0, \ldots, k$). However, there are several optimizations we can make. Algorithm 2 can be

implemented in a way that the $L_\ell^i$'s are not directly stored at all. To avoid storing the $L_\ell^i$'s, we can augment $H$ to contain not just $e^\star$, but also the index of the window it was added in. The index of the window tells us the range of levels that $e^\star$ was inserted into, so all of the $L_\ell^i$'s can be reconstructed from $H$ as $H$ contains a history of all the insertions. Thus the memory use of Algorithm 2 is the size of $H$ at the end of the stream. Since there are $\alpha k$ windows and each window introduces at most 1 element to $H$, we have the following observation:

**Observation 2.1.** *Algorithm 2 uses at most $O(\alpha k)$ space and $O(n\alpha\sqrt{k\log k})$ time.*

When $E$ is streamed in random order, our partitioning procedure (Algorithm 1) has a much simpler interpretation. (A similar lemma can be found in the appendix of Agrawal et al. [ASS19].)

**Lemma 2.1.** *Suppose $E$ is streamed according to a permutation chosen at random and we partition $E$ by Algorithm 1 into $m$ windows. This is equivalent to assigning each $e \in E$ to one of $m$ different buckets uniformly and independently at random.*

The algorithm's performance and behavior depends on the ordering of $E$. Let us define the *history* of the algorithm up to the $i$-th window, denoted $\mathcal{H}_i$, to be the sequence of all solutions produced up to that point. (Note that this history is only used in the analysis.) More precisely, we define the history as follows.

**Definition 2.1.** *Let $H_i$ denote the state of the set $H$ maintained by the algorithm, before processing window $i$. We define $\mathcal{H}_i$ to be the set of all triples $(e, \ell, j)$ such that element $e \in H_i$ was added to solution $L_\ell$ in window $j$. In other words, $\mathcal{H}_i$ contains all of the changes that the algorithm made to its state while processing the first $i$ windows. For convenience, sometimes we treat $\mathcal{H}_i$ as a set of elements and say that $e \in \mathcal{H}_i$ if $e \in H_i$.*

The history $\mathcal{H}_i$ describes the entire memory state of the algorithm up to the end of window $i-1$. In the following, we analyze the performance of the algorithm in the $i$-th window conditioned the history $\mathcal{H}_i$. Note that different random permutations of the input may produce this history, and we average over all of them in the analysis.

The next key lemma captures the intuition that elements not selected by the algorithm so far could still appear in the future, and bounds the probability with which this happens.

**Lemma 2.2.** *Fix a history $\mathcal{H}_{i-1}$. For any element $e \in E \setminus \mathcal{H}_{i-1}$, and any $i \leq i' \leq m = \alpha k$, we have $\Pr\left(e \in w_i' \mid \mathcal{H}_{i-1}\right) \geq 1/(\alpha k)$.*

Next, we define a set of *active* windows. In each active window, the algorithm is expected to make significant improvements to its candidate solution. The active windows will only be used in the analysis of the algorithm, and need not be computed in any way.

**Definition 2.2.** *Let $O$ be the optimal solution. For window $i$, let $p_e^i$ be the probability that $e \in w_i$ given $\mathcal{H}_{i-1}$. Define its active set $A_i$ to be the union of $R_i$ and the set obtained by sampling each $e \in w_i$ with probability $1/(\alpha k p_e^i)$. We call $w_i$ an **active window** if $|O \cap A_i| \geq 1$ and we call $O \cap A_i$ the **active optimal elements** of window $i$.*

Note that the construction of active sets in Definition 2.2 is valid as Lemma 2.2 guarantees $1/(\alpha k p_e^i) \leq 1$. More importantly, the active window $A_i$ subsamples the optimal elements so that each element appears in $A_i$ with probability exactly $1/(\alpha k)$ regardless of the history $\mathcal{H}_{i-1}$. This allows us to tightly bound the number of active windows in the input, as we show in the next lemma.

**Lemma 2.3.** *Suppose we have streamed up to the $\alpha\beta$-th window of the input for some $\beta > 0$. Then expected number of active windows seen so far satisfies*

$$\bar{Z}_{\alpha\beta} := \text{expected number of active windows} = \beta - \Theta(\beta/\alpha).$$

*Furthermore, the actual number of windows concentrates around $\bar{Z}_{\alpha\beta}$ to within $\pm O\left(\sqrt{\beta \log \frac{1}{\delta}}\right)$ with probability $1 - \delta$.*

Next we analyze the expected gain in the solution after processing each active window. Let $\mathcal{A}_{i+1}$ the event that window $w_{i+1}$ is active.

**Lemma 2.4.** *Let $\bar{\mathcal{L}}_i = \{z_l^i, z_l^i + 1, \ldots, z_h^i\}$ where $z_l^i$ and $z_h^i$ are the values of $z_l$ and $z_h$ defined in Algorithm 2 on window $i$. Conditioned on a history $\mathcal{H}_i$ and window $i+1$ being active,*

$$\sum_{\ell \in \bar{\mathcal{L}}_{i+1}} \mathbf{E}[f(L_{\ell+1}^{i+1}) - f(L_\ell^i) \mid \mathcal{H}_i, \mathcal{A}_{i+1}] \geq \frac{1}{k} \sum_{\ell \in \bar{\mathcal{L}}_{i+1}} (f(O) - \mathbf{E}[f(L_\ell^i) \mid \mathcal{H}_i]). \tag{1}$$

Under an ideal partition, each element of $O$ appears in a different window, with one optimal element appearing roughly once every $\alpha$ windows. Thus after $k$ active windows, we expect to obtain a $1 - (1 - 1/k)^k \approx (1 - 1/e)$-approximation (as in the standard greedy analysis).

**Theorem 2.5.** *The expected value of the best solution found by the algorithm is at least*

$$\left(1 - \frac{1}{e} - O\left(\frac{1}{\alpha} + \alpha\sqrt{\frac{\log k}{k}}\right)\right) OPT.$$

*Setting $\alpha = \Theta(1/\varepsilon)$, we have a $(1 - 1/e - \varepsilon - o(1))$-approximation using $O(k/\varepsilon)$ memory.*

# 3  A $(1/e - \varepsilon)$-approximation for non-monotone submodular maximization

In this section, we show that the basic algorithm described in Algorithm 2 can be altered to give a $1/e$-approximation to the cardinality constrained non-monotone case (Algorithm 3).

---

**Algorithm 3** NONMONOTONESTREAM$(f, E, k, \alpha)$

---

1: Partition $E$ into windows $w_i$ for $i = 1, \ldots, \alpha k$ with Algorithm 1.
2: $L_\ell^0 \leftarrow \emptyset$ for $i = 0, \ldots, k$
3: $H \leftarrow \emptyset$
4: $x_e^i \leftarrow \mathrm{Unif}(0, 1)$ for $e \in E, i \in [\alpha k]$
5: **for** $i = 1, \ldots, \alpha k$ **do**
6:     $C_i \leftarrow \emptyset$
7:     **for** $e \in$ window $w_i$ **do**
8:         **for** $j = 1, \ldots, i$ **do**
9:             Reconstruct $L_\ell^{j-1}$ for all $\ell$ from $\mathcal{H}_{i-1}$
10:            $A_e^j = \{r \mid 1 \leq r < j, \sum_{\ell=z_l^r}^{z_h^r} f(e|L_\ell^{r-1}) < f_r \text{ or } x_e^r > q_e^r\}$ (see line 13 for $f_r$)
11:            $q_e^j = \frac{\alpha k - j + |A_e^j| + 1}{\alpha k}$
12:        **if** $x_e^i \leq q_e^i$ **then** $C_i \leftarrow C_i \cup \{e\}$
13:     $f_i \leftarrow \sum_{\ell=z_i^i}^{z_h^i} f(e^\star | L_\ell^{i-1})$
14:     Perform lines 7 to 16 of Algorithm 2.
15: **return** $\arg\max_\ell f(L_\ell^{\alpha k})$

---

Algorithm 3 uses the same kind of multi-level scheme as Algorithm 2. However, Algorithm 3 further sub-samples the elements of the input so that the probability of including any element is *exactly* $1/(\alpha k)$ lines 7–13 (coloured in orange). The sub-sampling allows us to bound the maximum probability that an element of the input is included in the solution. In particular, the sub-sampling is done by having the algorithm compute (on the fly) the conditional probability that an element $e$ could have been selected had it appeared in the past. This gives us the ability to compute an appropriate sub-sampling probability to ensure that $e$ does not appear in $H$ with too high a probability. In terms of the proof, the sub-sampling allows us to perform a similar analysis to the RANDOMGREEDY algorithm of Buchbinder et al. [BFNS14].[3]

Since many of the main ideas are the same, we relegate the details of the analysis to the supplementary.

**Implementation of Algorithm 3**  For clarity of exposition, we compute $x_e^i$ up front in line 4. However, we can compute them on the fly in practice since each element only uses its value of $x_e^i$ once (lines 10 and 12). This avoids an $O(n\alpha k)$ memory cost associated with storing each $x_e^i$. Finally, we assume that there are no ties when computing the best candidate element in each window. Ties can be handled by any arbitrary but consistent tie-breaking procedure. Any additional information used to break the ties (for example an ordering on the elements $e$) must be stored alongside $f_i$ for the computation of $A_e^j$ (line 10).

**Theorem 3.1.** *Algorithm 3 obtains a $(1/e - \varepsilon)$-approximation for maximizing a non-monotone function $f$ with respect to a cardinality constraint in $O(k/\varepsilon)$ memory.*

---

[3]A difference here is that instead of analysing a random element of the top-$k$ marginals, we analyse the optimal set directly.

We remark that Algorithm 3 also achieves a guarantee of $1 - 1/e - \varepsilon$ for the monotone case, as Lemma 2.4 and Theorem 2.5 both still apply to Algorithm 3 when $f$ is monotone. The main difference between the two is the sub-sampling (lines 7–13), which increases the running time of the algorithm.

## 4 $1 - 1/e$ hardness for monotone submodular maximization

The proofs of the following propositions and lemmas may be found in the supplementary.

**Proposition 4.1.** *Fix subsets $G, B$ of elements (denoting "good" and "bad") such that $|G| = k$ and $|B| = n - k$; let $r \in [0, k]$ be some parameter. Let $m$ denote the size of the memory buffer, and let $p$ denote the probability that a random subset of size $m$ contains at least $r - 1$ good elements. Let $f : G \cup B \to \mathbb{R}$ be a function that satisfies the following symmetries:*

- *$f$ is symmetric over good (resp. bad) elements, namely there exists $\hat{f}$ such that $f(S) = \hat{f}(|S \cap G|, |S \cap B|)$.*

- *For any set $S$ with $\leq r - 1$ good elements, $f$ does not distinguish between good and bad elements, namely for $g \leq r - 1$, $\hat{f}(g, b) = \hat{f}(0, b + g)$.*

*Then any algorithm has expected value at most*

$$ALG \leq (1 - pk)\hat{f}(0, k) + pk \cdot OPT. \tag{2}$$

We now consider a few different $f$'s that satisfy the desiderata of Proposition 4.1.

**Lemma 4.1** (monotone submodular function [NW78])**.** *There exists a monotone submodular $f$ that satisfies the desiderata of Proposition 4.1 for $r = 2\varepsilon k$, and such that:*

- *$\hat{f}(0, k) = 1 - 1/e + O(\varepsilon)$.*

- *$OPT = f(G) = 1$.*

**Lemma 4.2** (polynomial-universe coverage function [MV19])**.** *There exists a (monotone submodular) coverage function $f$ over a polynomial universe $U$ that satisfies the desiderata of Proposition 4.1 for $r = 2$, and such that:*

- *$\hat{f}(0, k) = (1 - 1/e + o(1))|U|$.*

- *$OPT = f(G) = |U|$.*

**Lemma 4.3** (exponential-universe coverage function (**new construction**))**.** *There exists a (monotone submodular) coverage function $f$ over an exponential universe $U$ that satisfies the desiderata of Proposition 4.1 for $r = 3$, and such that:*

- *$\hat{f}(0, k) = (1 - 1/e + o(1))|U|$.*

- *$OPT = f(G) = |U|$.*

Our main hardness result follows from the lemmas above:

**Theorem 4.4.** *Any $(1 - 1/e + \varepsilon)$-approximation algorithm in the random order strong oracle model must use the following memory:*

- *$\Omega(n)$ for a general monotone submodular function.*

- *$\Omega(n/k^2)$ for a coverage function over a polynomial universe.*

- *$\Omega(n/k^{3/2})$ for a coverage function over an exponential universe.*

## 5 Experimental results

In the following section, we give experimental results for our monotone streaming algorithm. Due to space limitations, the experiments for the non-monotone algorithm can be found in the supplementary.

Our main goal is to show that our algorithm performs well in a practical setting and is simple to implement. In fact, we show that our algorithm is on par with *offline* algorithms in performance, and returns competitive solutions across a variety of datasets. All experiments were performed on a 2.7 GHz dual-core Intel i7 CPU with 16 GB of RAM.

We compare the approximation ratios obtained by our algorithm with three benchmarks:

- The *offline* LAZYGREEDY algorithm [Min78], which is both theoretically optimal and obtains the same solution as greedy (in faster time). Note that we don't expect to outperform it with a streaming algorithm; but as we hoped, our algorithm comes close.

- The SIEVESTREAMING algorithm of Badanidiyuru et al. [BMKK14], which is the first algorithm to appear for adversarial streaming submodular optimization.

- The SALSA algorithm of Norouzi-Fard et al. [NTM+18], which is the first "beyond 1/2" approximation algorithm for random-order streams. This algorithm runs several varients of SIEVESTREAMING in parallel with thresholds that change as the algorithm progresses through the stream. [4] for *adversarial* order streaming. As we would expect, our algorithm performs better on random arrival streams.

Note that in terms of memory use, our algorithm is strictly more efficient. The analysis in previous sections show that the memory is $O(k/\varepsilon)$ (with a small constant), versus $O(k \log k/\varepsilon)$ for both SIEVESTREAMING and SALSA. Thus in the experiments below, we focus on the approximation ratio obtained by our algorithm.

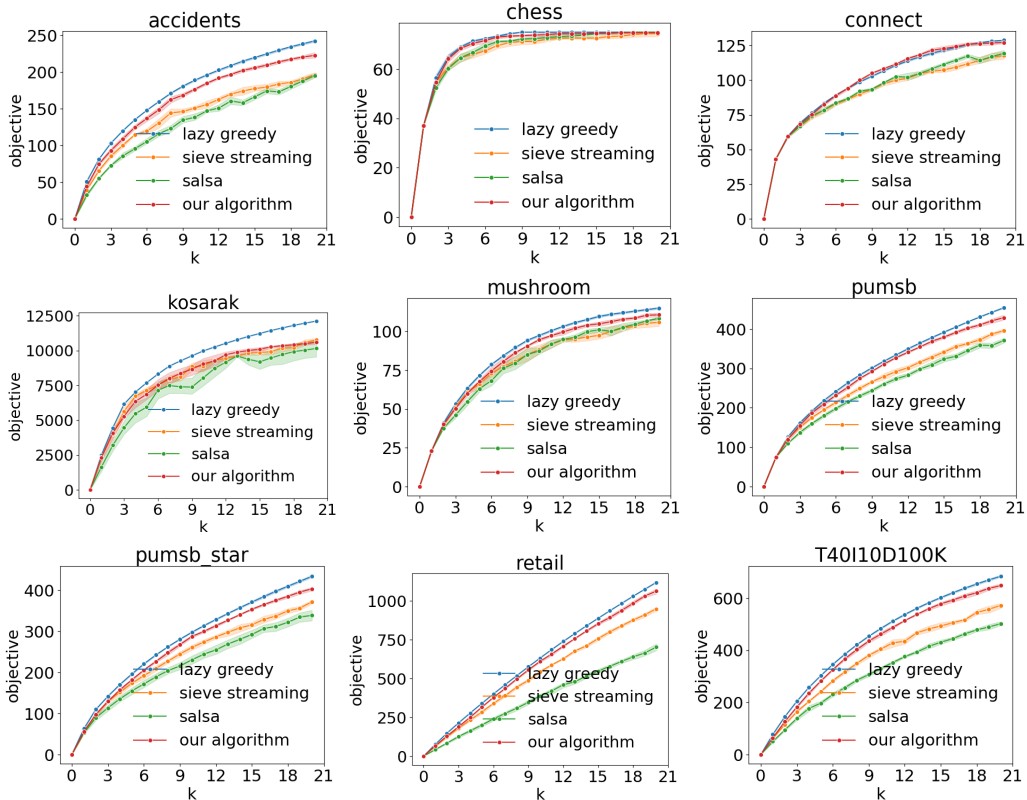

Figure 1: Performance of lazy greedy, sieve streaming, and our algorithm on each data set (averaged across 10 runs, shaded regions represent variance across different random orderings).

---

[4]SIEVESTREAMING is also known as threshold greedy in the literature [BV14]. Note that the later SIEVESTREAMING++ algorithm of [KMZ+19] is more efficient, but for approximation ratio SIEVESTREAMING is a stronger benchmark.

**Datasets** Our datasets are drawn from set coverage instances from the 2003 and 2004 workshops on Frequent Itemset Mining Implementations [oDM04] and the Steiner triple instances of Beasley [Bea87]. For each data set we run the three algorithms for cardinality constraints varying from $k = 1$ to 20. The results of the algorithms are averaged across 10 random stream orderings. Table 2 describes the data sources. Figure 1 shows the performance of the three algorithms on each data set. All code can be found at `https://github.com/where-is-paul/submodular-streaming` and all datasets can be found at `https://tinyurl.com/neurips-21`.

| dataset | source | # of sets | universe size |
|---|---|---|---|
| accidents | (anonymized) traffic accident data | 340183 | 468 |
| chess | UCI ML Repository | 3196 | 75 |
| connect | UCI ML Repository | 67557 | 129 |
| kosarak | (anonymized) click-stream data | 990002 | 41270 |
| mushroom | UCI ML Repository | 8124 | 119 |
| pumsb | census data for population and housing | 49046 | 7116 |
| pumsb_star | census data for population and housing | 49046 | 7116 |
| retail | (anonymized) retail market basket data | 88162 | 16469 |
| T40I10D100K | generator from IBM Quest research | 100000 | 999 |

Table 2: Description of data sources for monotone experiments.

## 6 Conclusion and Future Work

In this work, we have presented memory-optimal algorithms for the problem maximizing submodular functions with respect to cardinality constraints in the random order streaming model. Our algorithms achieve an optimal approximation factor of $1 - 1/e$ for the monotone submodular case, and an approximation factor of $1/e$ for the non-monotone case. In addition to theoretical guarantees, we show that the algorithm outperforms existing state-of-the-art on a variety of datasets.

We close with a few open questions that would make for interesting future work. Although our algorithm is memory-optimal, it is not runtime-optimal. In particular, the SIEVESTREAMING [BMKK14] and SALSA [NTM+18] algorithms both run in time $O(n \log k/\varepsilon)$, whereas our algorithm runs in time $O(n\sqrt{k \log k}/\varepsilon)$. The non-monotone variant of our algorithm runs even slower, as it needs to perform sub-sampling operations that take at least $O(\alpha^3 k^{5/2})$ per stream element in its current form. Improving this runtime would greatly improve the practicality of our algorithm for extremely large cardinality constraints. Finally, there has been recent interest in examining streaming algorithms for streams under "adversarial injection" [GKRS20]. In such streams, the optimal elements of the stream are randomly ordered, while adversarial elements can be injected between the optimal elements with no constraints. Despite the seemingly large power of the adversary, the $1/2$ approximation barrier can still be broken in this model. It would be interesting to see if the work in this paper can be extended to such a setting.

## Acknowledgments and Disclosure of Funding

The authors are indebted to Mohammad Shadravan and Morteza Monemizadeh for their insightful discussions that no doubt improved this work. The first author is supported by a VMWare Fellowship and the Natural Sciences and Engineering Research Council of Canada. The second and fourth authors are supported by NSF CCF-1954927, and the second author is additionally supported by a David and Lucile Packard Fellowship.

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
