## Supplementary

## A  Missing proofs from Section 2

We begin with a simple lemma showing that the values of the levels are *monotone*:

**Lemma A.1.** *For all $i$ and $\ell$, $f(L_\ell^i) \leq f(L_{\ell+1}^{i+1})$ and $f(L_\ell^i) \leq f(L_{\ell+1}^i)$.*

*Proof.* First, we note that the second part of the lemma holds by lines 15–16. Let $z_l^i$ and $z_h^i$ be the value of $z_l$ and $z_h$ in Algorithm 2 on line 9 on window $i$. Consider a window $i+1$. There are two cases, depending on whether an element $e^\star$ was added to the solutions or not. Suppose no element $e^\star$ was added to the solution. Then all the levels remain the same. Line 15 guarantees that $f(L_\ell^i) \leq f(L_{\ell+1}^i)$. Since no elements were added, $f(L_{\ell+1}^i) = f(L_{\ell+1}^{i+1})$ so $f(L_\ell^i) \leq f(L_{\ell+1}^{i+1})$ for every level $\ell$. Now suppose an element $e^\star$ was added in window $i+1$. For levels $\ell > z_h^{i+1}$ and $\ell < z_l^{i+1}$, $L_{\ell+1}^{i+1} = L_{\ell+1}^i$, so $f(L_{\ell+1}^{i+1}) = f(L_{\ell+1}^i) \geq f(L_\ell^i)$. For levels $\ell \in [z_l^{i+1}, z_h^{i+1}]$, $L_{\ell+1}^{i+1} = L_\ell^i \cup \{e^\star\}$ at the end of line 13, and its value only improves through lines 15–16. Thus $f(L_{\ell+1}^{i+1}) \geq f(L_\ell^i \cup \{e^\star\}) \geq f(L_\ell^i)$.  $\square$

The rest of the proofs below correspond directly to unproven lemmas in Section 2.

**Lemma 2.1.** *Suppose $E$ is streamed according to a permutation chosen at random and we partition $E$ by Algorithm 1 into $m$ windows. This is equivalent to assigning each $e \in E$ to one of $m$ different buckets uniformly and independently at random.*

*Proof.* The way we define the window sizes is equivalent to placing each element independently into a random bucket, and letting $n_i$ be the number of elements in bucket $i$. Hence the distribution of window sizes is correct. Conditioned on the window sizes, the assignment of elements into windows is determined by a random permutation; any partition is equally likely. Therefore the distribution of elements into windows is equivalent to placing each each element into a random window independently.  $\square$

**Definition A.1.** *Let $P : E \to [\alpha k]$ be the mapping of $E$ to their window indices; i.e. if $e$ is in the $j$-th window, then $P(e) = j$. A partition $P$ is $\mathcal{H}_i$-**compatible** if the algorithm produces history $\mathcal{H}_i$ when streaming the first $i$ windows partitioned by $P$.*

**Lemma 2.2.** *Fix a history $\mathcal{H}_{i-1}$. For any element $e \in E \setminus \mathcal{H}_{i-1}$, and any $i \leq i' \leq m = \alpha k$, we have $\Pr\left(e \in w_i' \mid \mathcal{H}_{i-1}\right) \geq 1/(\alpha k)$.*

*Proof.* Let $e$ be any element of $E \setminus \mathcal{H}_{i-1}$ and choose any $j, j' \in [m]$. For each $\mathcal{H}_{i-1}$-compatible partition $P$ with $P(e) = j'$, we begin by showing that we can create another $\mathcal{H}_{i-1}$-compatible partition $\tilde{P}$ by setting $\tilde{P}(e) = j$ and all other values of $\tilde{P}$ equal to $P$. In other words, any $\mathcal{H}_{i-1}$-compatible partition where $e$ is in window $j$ can be mapped to another where $e$ is in window $j'$.

Observe that because $e \notin \mathcal{H}_{i-1}$, $e$ must not have been chosen in windows $j$ or $j'$ by the algorithm so far. There are two possible reasons for this: first, windows $j$ or $j'$ could be greater than (further in the future) or equal to window $i$, in which case window $j, j'$ trivially does not affect $\mathcal{H}_{i-1}$. Second, consider the case when windows $j$ or $j'$ is less than $i$. If this is the case, then element $e$ already arrived in the stream but was not selected by the algorithm in any solution. Hence $e$ was either never the maximum element found in line 10, or if it was, its marginal value was not sufficient to replace the current solution. In either case, removing or adding $e$ to windows $j$ and $j'$ will not change the history $\mathcal{H}_{i-1}$: if a different element $e'$ was chosen for the update in window $j'$, this will still be the case; and if no update occurred, this will also still be the case. Finally, observe that the pool of elements $H$ for re-insertion will also not be affected, since element $e$ was not part of it.

Thus, we may change $P(e)$ from $j'$ to $j$ and maintain a $\mathcal{H}_{i-1}$-compatible partition. Since $\tilde{P}$ is equal to $P$ everywhere except on $e$, this maps each such partition $P$ to a unique partition $\tilde{P}$ (and vice versa), establishing a bijectiion between $\mathcal{H}_{i-1}$-compatible partitions with $e \in w_j$ and $e \in w_{j'}$. Consequently, the number of partitions with $P(e) = j$ compatible with $\mathcal{H}_{i-1}$ is equal to the number of partitions with $P(e) = j'$.

Let $J_e$ be the set of indices $j$ where there exists $\mathcal{H}_{i-1}$-compatible partitions with $e \in w_j$. The argument above applies to any windows $j \in J_e$. In particular, $J_e$ contains all windows greater than or equal to $i$, since these windows clearly do not affect $\mathcal{H}_{i-1}$. For any element $e \notin \mathcal{H}_{i-1}$, and $j, j' \in J_e$, we have

$$
\begin{aligned}
\Pr\left(e \in w_j \mid \mathcal{H}_{i-1}\right) &= \frac{\#\text{partitions } P \text{ with } P(e) = j \text{ and } P \text{ is } \mathcal{H}_{i-1}\text{-compatible}}{\#\text{partitions } P \text{ where } P \text{ is } \mathcal{H}_{i-1}\text{-compatible}} \\
&= \frac{\#\text{partitions } P \text{ with } P(e) = j' \text{ and } P \text{ is } \mathcal{H}_{i-1}\text{-compatible}}{\#\text{partitions } P \text{ where } P \text{ is } \mathcal{H}_{i-1}\text{-compatible}} \\
&= \Pr\left(e \in w_{j'} \mid \mathcal{H}_{i-1}\right).
\end{aligned}
$$

The first and last lines follow from Lemma 2.1, since any partition happens with uniform probability.

Any element $e \notin \mathcal{H}_{i-1}$ must appear in some window, and it is equally likely to be in any of the windows where it could be present without affecting the history $\mathcal{H}_{i-1}$. So we have

$$
1 = \sum_{s=1}^{\alpha k} \Pr\left(e \in w_s \mid \mathcal{H}_{i-1}\right) = |J_e| \cdot \Pr\left(e \in w_i \mid \mathcal{H}_{i-1}\right)
$$

The lemma follows from noting that $|J_e| \le \alpha k$. $\qquad\square$

*Proof.* By the definition of an active set, $\Pr\left(\mathbb{1}_{o \in A_i} \mid \mathcal{H}_{i-1}\right) = 1/(\alpha k)$ for any $\mathcal{H}_{i-1}$ and $o \in O \setminus \mathcal{H}_{i-1}$. For $o \in \mathcal{H}_{i-1} \cap O$, these elements are reintroduced by the algorithm with probability $1/(\alpha k)$. Hence, $\Pr\left(\mathbb{1}_{o \in A_i}\right) = 1/(\alpha k)$ for any $o \in O$, without conditioning on $\mathcal{H}_{i-1}$. Since the input permutation is uniformly random, $\mathbb{1}_{o_a \in A_i}$ and $\mathbb{1}_{o_b \in A_i}$ are independent for $a \ne b$ in any window $i$. Letting $\mathbb{1}_i = \cup_{o \in O} \mathbb{1}_{o_a \in A_i}$, we have

$$
\Pr\left(\mathbb{1}_i\right) = 1 - \left(1 - \frac{1}{\alpha k}\right)^k = 1/\alpha - 1/(2\alpha^2) + O(1/\alpha^3)
$$

for large enough $\alpha$ and $k$. Thus,

$$
\mathbf{E} \sum_{i=1}^{\alpha\beta} \mathbb{1}_i = \alpha\beta \cdot \mathbf{E}\mathbb{1}_1 = (1 - 1/(2\alpha) + O(1/\alpha^2))\beta.
$$

Next, note that $\mathbb{1}_i$ and $\mathbb{1}_j$ are negatively dependent: conditioning on a window being active decreases the number of optimal elements available to the other windows (and conditioning on a window not being active increases the number available to other windows). Thus we have:

$$
\begin{aligned}
\mathbf{Var}\left(\sum \mathbb{1}_i\right) &\le \alpha\beta(\mathbf{E}\mathbb{1}_1 - \mathbf{E}\mathbb{1}_1{}^2) \\
&= (1 - 3/(2\alpha) + O(1/\alpha^2))\beta.
\end{aligned}
$$

Now can apply the lower-tail bound Hoeffding's inequality ([Doe20], Theorem 1.10.12) to get

$$
\begin{aligned}
\Pr\left(\sum_{i=1}^{\alpha\beta} \mathbb{1}_i \le -c\sqrt{\beta \log \frac{1}{\delta}} + \mathbf{E} \sum_{i=1}^{\alpha\beta} \mathbb{1}_i\right) &\le \exp\left(-\frac{c^2 \log \frac{1}{\delta}}{3}\right) \\
&\le \delta/2
\end{aligned}
$$

for a large enough constant $c$.

Similarly, we may apply the upper-tail bound of Hoeffding's inequality [Doe20] (Corollary 1.10.13), to obtain:

$$
\Pr\left(\sum_{i=1}^{\alpha\beta} \mathbb{1}_i \le c\sqrt{\beta \log \frac{1}{\delta}} + \mathbf{E} \sum_{i=1}^{\alpha\beta} \mathbb{1}_i\right) \le \delta/2.
$$

for a large enough constant $c$.

Since $\mathbf{E} \sum_{i=1}^{\alpha\beta} \mathbb{1}_i = \beta - \Theta(\beta/\alpha)$, the number of active windows in the first $\alpha\beta$ windows is at least $\beta - O\left(\sqrt{\beta \log \frac{1}{\delta}}\right) - \Theta(\beta/\alpha)$ and at most $\beta + O\left(\sqrt{\beta \log \frac{1}{\delta}}\right) - \Theta(\beta/\alpha)$ with probability at least $1 - \delta$. $\qquad\square$

**Lemma 2.3.** *Suppose we have streamed up to the $\alpha\beta$-th window of the input for some $\beta > 0$. Then expected number of active windows seen so far satisfies*

$$\bar{Z}_{\alpha\beta} := \text{expected number of active windows} = \beta - \Theta(\beta/\alpha).$$

*Furthermore, the actual number of windows concentrates around $\bar{Z}_{\alpha\beta}$ to within $\pm O\left(\sqrt{\beta \log \frac{1}{\delta}}\right)$ with probability $1 - \delta$.*

**Lemma 2.4.** *Let $\bar{\mathcal{L}}_i = \{z_l^i, z_l^i + 1, \ldots, z_h^i\}$ where $z_l^i$ and $z_h^i$ are the values of $z_l$ and $z_h$ defined in Algorithm 2 on window $i$. Conditioned on a history $\mathcal{H}_i$ and window $i + 1$ being active,*

$$\sum_{\ell \in \bar{\mathcal{L}}_{i+1}} \mathbf{E}[f(L_{\ell+1}^{i+1}) - f(L_\ell^i) \mid \mathcal{H}_i, \mathcal{A}_{i+1}] \geq \frac{1}{k} \sum_{\ell \in \bar{\mathcal{L}}_{i+1}} (f(O) - \mathbf{E}[f(L_\ell^i) \mid \mathcal{H}_i]). \quad (3)$$

*Proof.* As in the previous lemma, we first note that by the construction of the active set $A_{i+1}$, any $e \in E$ appears in $A_{i+1}$ with probability exactly $1/(\alpha k)$, so $p := \Pr(o_j \in A_{i+1} | \mathcal{H}_i) = 1/(\alpha k)$ for any $o_j \in O$. Also, the appearances of different elements are mutually independent. In particular, we have $\Pr(\mathcal{A}_{i+1} \mid \mathcal{H}_i) = \Pr(\exists o \in O \cap A_{i+1} \mid \mathcal{H}_i) = 1 - (1 - p)^k$.

Order the $o_j$'s so that $\sum_{\ell \in \bar{\mathcal{L}}_{i+1}} f(o_j | L_\ell^i) \geq \sum_{\ell \in \bar{\mathcal{L}}_{i+1}} f(o_{j'} | L_\ell^i)$ for $j \leq j'$. Given that $o \in A_{i+1}$ for some random $o \in O$, we have

$$\sum_{\ell \in \bar{\mathcal{L}}_{i+1}} \mathbf{E}[f(L_{\ell+1}^{i+1}) - f(L_\ell^i) \mid \mathcal{H}_i, \mathcal{A}_{i+1}] \geq \max_{e \in A_{i+1}} \sum_{\ell \in \bar{\mathcal{L}}_{i+1}} \mathbf{E}\left[f(e | L_\ell^i) \mid \mathcal{H}_i, \mathcal{A}_{i+1}\right]$$

$$\geq \sum_{j=1}^k \frac{p(1-p)^{j-1}}{1 - (1-p)^k} \sum_{\ell \in \bar{\mathcal{L}}_{i+1}} \mathbf{E}\left[f(o_j | L_\ell^i) \mid \mathcal{H}_i, \mathcal{A}_{i+1}\right]$$

$$\geq \sum_{\ell \in \bar{\mathcal{L}}_{i+1}} \mathbf{E}\left[\frac{1}{k} \sum_{j=1}^k f(o_j | L_\ell^i) \mid \mathcal{H}_i, \mathcal{A}_{i+1}\right]$$

$$\geq \sum_{\ell \in \bar{\mathcal{L}}_{i+1}} \mathbf{E}\left[\frac{1}{k}(f(O \cup L_\ell^i) - f(L_\ell^i)) \mid \mathcal{H}_i, \mathcal{A}_{i+1}\right] \quad \text{(Submodularity)}$$

$$\geq \sum_{\ell \in \bar{\mathcal{L}}_{i+1}} \mathbf{E}\left[\frac{1}{k}(f(O) - f(L_\ell^i)) \mid \mathcal{H}_i, \mathcal{A}_{i+1}\right] \quad \text{(Monotonicity)}$$

$$\geq \sum_{\ell \in \bar{\mathcal{L}}_{i+1}} \mathbf{E}\left[\frac{1}{k}(f(O) - f(L_\ell^i)) \mid \mathcal{H}_i\right]. \quad (4)$$

The first line follows from the fact that $A_{i+1} \subseteq C_{i+1}$, since $C_{i+1}$ contains $R_{i+1}$ and the entirety of $w_{i+1}$. The numerator of $\frac{p(1-p)^{j-1}}{1-(1-p)^k}$ follows from the fact that if $o_j$ is the maximum, then no elements in $O$ valued higher than it can appear in $A_{i+1}$. The denominator is the probability that window $w_{i+1}$ is active. (All events are conditioned on $\mathcal{H}_i$.) The third line is subtle and follows from Chebyshev's sum inequality. Let $\alpha_j = \frac{p(1-p)^{j-1}}{1-(1-p)^k}$ and $\beta_j = \sum_{\ell \in \bar{\mathcal{L}}_{i+1}} \mathbf{E}[f(o_j | L_\ell^i) \mid \mathcal{H}_i, \mathcal{A}_{i+1}]$. Since $\alpha_j$ and $\beta_j$ are both decreasing sequences, Chebyshev's sum inequality gives:

$$\sum_{j=1}^k \alpha_j \beta_j \geq \frac{1}{k} \sum_{j=1}^k \alpha_j \sum_{j=1}^k \beta_j = \frac{1}{k} \sum_{j=1}^k \sum_{\ell \in \bar{\mathcal{L}}_{i+1}} \mathbf{E}[f(o_j | L_\ell^i) \mid \mathcal{H}_i, \mathcal{A}_{i+1}]. \quad \square$$

Finally, note that $\mathcal{H}_i$ is independent of $\mathcal{A}_{i+1}$, as $\Pr(\mathcal{A}_{i+1} \mid \mathcal{H}_i)$ is constructed to be the same regardless of $\mathcal{H}_i$. Hence the conditioning on $\mathcal{A}_{i+1}$ can be removed due to the independence and the fact that $L_\ell^i$ only depends on $\mathcal{H}_i$ and not $\mathcal{A}_{i+1}$.

**Theorem 2.5.** *The expected value of the best solution found by the algorithm is at least*

$$\left(1 - \frac{1}{e} - O\left(\frac{1}{\alpha} + \alpha\sqrt{\frac{\log k}{k}}\right)\right) OPT.$$

*Setting $\alpha = \Theta(1/\varepsilon)$, we have a $(1 - 1/e - \varepsilon - o(1))$-approximation using $O(k/\varepsilon)$ memory.*

*Proof.* Let $Z_i = \sum_{j=1}^{i} \mathbb{1}_{\mathcal{A}_j}$ be a random variable which indicates the number of active windows up to window $i$. Recall from Lemma 2.3 that $\bar{Z}_i := \mathbf{E}[Z_i] = i/\alpha - \Theta(i/\alpha^2)$, and furthermore $Z_i$ concentrates around its expectation: For $\sigma := 20\sqrt{k \log k}$, we have that w.h.p. $(1 - \mathrm{poly}(1/k))$,

$$\forall i \quad Z_i \in [\bar{Z}_i \pm \sigma]. \tag{5}$$

By the definition of $O$, $f(L_\ell^i) \leq f(O)$ for any $\ell$ and $i$ since $|L_\ell^i| \leq k$. Since (5) fails only with probability $\mathrm{poly}(1/k)$, we can condition on it while only having negligible effect of $\pm\mathrm{poly}(1/k)f(O)$ on $\mathbf{E}[f(L_\ell^i)]$ for any $i, \ell$. We will henceforth simply assume that it always holds; the (in)equalities for the rest of the proof of this lemma hold up to this $\pm\mathrm{poly}(1/k)f(O)$ term.

Our goal is to use Lemma 2.4 to argue about the progress of $\mathbf{E}[f(L_\ell^i)]$ for $\ell = Z_i$. However, we don't know how to prove an analogue of Equation (1) that also conditions on $\ell = Z_i$. We circumvent this issue by tracking the average of $\mathbf{E}[f(L_\ell^i)]$ over $\ell$ in an interval $\mathcal{L}_i$ around $Z_i$.

Let $\bar{\mathcal{L}}_i := [\bar{Z}_i \pm \alpha\sigma]$ and $\mathcal{L}_i := [Z_i \pm (\alpha\sigma + \sigma)]$. To keep both nonnegative, we denote $i' := i - \alpha(\alpha\sigma + \sigma) - \Theta(1)$ and consider the contribution from $i$ s.t. $i' \geq 0$. By Equation (5), the interval $\mathcal{L}_i$ has a large overlap with an interval $\bar{\mathcal{L}}_i$ around $\bar{Z}_i$. Since $\bar{\mathcal{L}}_i$ does not depend on $Z_i$ (or the history in general), we can safely apply Lemma 2.4 for all $\ell \in \bar{\mathcal{L}}_i$.

We will argue that on average over $\ell \in \mathcal{L}_i$ and the randomness of the stream, $\mathbf{E}[f(L_\ell^i)]$ increases like

$$\frac{1}{|\mathcal{L}_i|} \sum_{\ell \in \mathcal{L}_i} \mathbf{E}[f(L_\ell^i)] \geq \left(1 - \left(1 - \frac{1}{k}(1 - e^{-1/\alpha})\right)^{i'} - O\left(\frac{i}{\alpha^2 k}\right)\right) f(O). \tag{6}$$

In particular, for $\tau := \alpha(k - \alpha\sigma - 2\sigma)$, we have that

$$\frac{1}{|\mathcal{L}_\tau|} \sum_{\ell \in \mathcal{L}_\tau} \mathbf{E}[f(L_\ell^\tau)] \geq \left(1 - 1/e - O\left(\frac{1}{\alpha} + \frac{\alpha\sigma}{k}\right)\right) f(O).$$

Assuming Equation (5), the maximum level in $\mathcal{L}_\tau$ is at most $Z_\tau + \alpha\sigma + \sigma \leq k$ w.h.p. The values of the levels are monotonically increasing due to line algorithm 2 and therefore level $k$ satisfies

$$\mathbf{E}\left[f(L_k^\tau)\right] \geq \left(1 - 1/e - O\left(\frac{1}{\alpha} + \frac{\alpha\sigma}{k}\right)\right) f(O).$$

We now prove Equation (6). For $i$ s.t. $i' \geq 0$, we have:

$$\frac{1}{|\mathcal{L}_{i+1}|} \sum_{\ell \in \mathcal{L}_{i+1}} \mathbf{E}[f(L_\ell^{i+1}) \mid \mathcal{H}_i, \mathcal{A}_{i+1}] = \frac{1}{|\mathcal{L}_i|} \sum_{\ell \in \mathcal{L}_i} \mathbf{E}[f(L_{\ell+1}^{i+1}) \mid \mathcal{H}_i, \mathcal{A}_{i+1}] \quad \text{(Def. of } \mathcal{L}_i)$$

$$= \frac{1}{|\mathcal{L}_i|} \sum_{\ell \in \mathcal{L}_i \setminus \bar{\mathcal{L}}_{i+1}} \mathbf{E}[f(L_{\ell+1}^{i+1}) \mid \mathcal{H}_i, \mathcal{A}_{i+1}] + \frac{1}{|\mathcal{L}_i|} \sum_{\ell' \in \bar{\mathcal{L}}_{i+1}} \mathbf{E}[f(L_{\ell'+1}^{i+1}) \mid \mathcal{H}_i, \mathcal{A}_{i+1}] \quad \text{(Equation (5))}$$

$$\geq \frac{1}{|\mathcal{L}_i|} \sum_{\ell \in \mathcal{L}_i \setminus \bar{\mathcal{L}}_{i+1}} \mathbf{E}[f(L_\ell^i) \mid \mathcal{H}_i, \mathcal{A}_{i+1}] + \frac{1}{|\mathcal{L}_i|} \sum_{\ell' \in \bar{\mathcal{L}}_{i+1}} \mathbf{E}[f(L_{\ell'+1}^{i+1}) \mid \mathcal{H}_i, \mathcal{A}_{i+1}] \quad \text{(Lemma A.1)}$$

$$\geq \frac{1}{|\mathcal{L}_i|} \sum_{\ell \in \mathcal{L}_i} \mathbf{E}[f(L_\ell^i) \mid \mathcal{H}_i, \mathcal{A}_{i+1}] + \frac{1}{|\mathcal{L}_i|} \sum_{\ell' \in \bar{\mathcal{L}}_{i+1}} \frac{1}{k}\mathbf{E}[f(O) - f(L_{\ell'}^i) \mid \mathcal{H}_i] \quad \text{(Lemma 2.4)}$$

$$= \frac{1}{|\mathcal{L}_i|} \sum_{\ell \in \mathcal{L}_i} \mathbf{E}[f(L_\ell^i) \mid \mathcal{H}_i] + \frac{1}{|\mathcal{L}_i|} \sum_{\ell' \in \bar{\mathcal{L}}_{i+1}} \frac{1}{k}\mathbf{E}[f(O) - f(L_{\ell'}^i) \mid \mathcal{H}_i] \quad (\mathcal{A}_{i+1} \text{ indep. } L_\ell^i \;\; \forall \ell)$$

$$\geq \frac{1}{|\mathcal{L}_i|} \sum_{\ell \in \mathcal{L}_i} \left(\frac{1}{k}f(O) + \frac{k-1}{k}\mathbf{E}[f(L_\ell^i) \mid \mathcal{H}_i]\right) - O\left(\frac{\sigma}{k|\mathcal{L}_i|}\right) f(O) \quad (|\mathcal{L}_i| - |\bar{\mathcal{L}}_{i+1}| = 2\sigma)$$

$$= \frac{1}{k}f(O) + \frac{k-1}{k}\left(\frac{1}{|\mathcal{L}_i|} \sum_{\ell \in \mathcal{L}_i} \mathbf{E}[f(L_\ell^i) \mid \mathcal{H}_i]\right) - O\left(\frac{\sigma}{k|\mathcal{L}_i|}\right) f(O).$$

The most subtle line of the argument is the one invoking the independence of $L_\ell^i$ and $\mathcal{A}_{i+1}$. This independence is true when conditioned on $\mathcal{H}_i$: $\mathcal{A}_{i+1}$ and $\mathcal{H}_i$ are independent[5] and $L_\ell^i$ is determined by $\mathcal{H}_i$.

---

[5]Recall that active sets are defined so that $\Pr(A_{i+1}|\mathcal{H}_i) = 1 - \left(1 - \frac{1}{\alpha k}\right)^k$ for any $\mathcal{H}_i$.

In order to avoid cluttering the notation below, every expectation below is conditioned on $\mathcal{H}_i$. To address the conditioning on $\mathcal{A}_{i+1}$, consider

$$\frac{1}{|\mathcal{L}_{i+1}|} \sum_{\ell \in \mathcal{L}_{i+1}} \mathbf{E}[f(L_\ell^{i+1})] = \frac{1 - \Pr[\mathcal{A}_{i+1}]}{|\mathcal{L}_{i+1}|} \sum_{\ell \in \mathcal{L}_{i+1}} \mathbf{E}[f(L_\ell^{i+1}) \mid \neg \mathcal{A}_{i+1}] + \frac{\Pr[\mathcal{A}_{i+1}]}{|\mathcal{L}_{i+1}|} \sum_{\ell \in \mathcal{L}_{i+1}} \mathbf{E}[f(L_\ell^{i+1}) \mid \mathcal{A}_{i+1}]$$

$$\geq \left(1 - \frac{\Pr[\mathcal{A}_{i+1}]}{k}\right) \frac{1}{|\mathcal{L}_i|} \sum_{\ell \in \mathcal{L}_i} \mathbf{E}[f(L_\ell^i)] + \frac{1}{k} \Pr[\mathcal{A}_{i+1}] f(O) - O\left(\frac{\sigma}{k|\mathcal{L}_i|}\right) \Pr[\mathcal{A}_{i+1}] f(O)$$

$$\geq \left(1 - \frac{1}{\alpha k}\right) \frac{1}{|\mathcal{L}_i|} \sum_{\ell \in \mathcal{L}_i} \mathbf{E}[f(L_\ell^i)] + \frac{1}{\alpha k} f(O) - O\left(\frac{\sigma}{\alpha k|\mathcal{L}_i|}\right) f(O).$$

where the last line comes from the inequality $1 - e^{-1/\alpha} \leq \Pr[\mathcal{A}_{i+1}] = 1 - (1 - \frac{1}{\alpha k})^k \leq \frac{1}{\alpha}$ and $\sum_{\ell \in \mathcal{L}_{i+1}} \mathbf{E}[f(L_\ell^{i+1}) \mid \neg \mathcal{A}_{i+1}] \geq \sum_{\ell \in \mathcal{L}_{i+1}} \mathbf{E}[f(L_\ell^i) \mid \neg \mathcal{A}_{i+1}]$ due to line 11 and the fact that $\bar{\mathcal{L}}_{i+1} \subseteq \mathcal{L}_i$ with high probability. Finally, given that $|\mathcal{L}_i| = 2(\alpha + 1)\sigma$, the last error term is $O\left(\frac{1}{\alpha^2 k}\right) f(O)$. Now we may remove the implicit conditioning on $\mathcal{H}_i$ (as we've conditioned everything on it so far). Equation (6) then follows by induction on $i$. $\qquad \square$

**Remark A.1.** *For any setting of $\varepsilon$, the approximation factor is at best $1 - \frac{1}{e} - \left(\frac{\log k}{k}\right)^{1/4}$, so we might as well choose $\varepsilon \geq \left(\frac{\log k}{k}\right)^{1/4}$. Agrawal et al. [ASS19] have a similar issue; the formal guarantee shown by Agrawal et al. is an approximation factor of $1 - 1/e - \varepsilon - \log(1/\varepsilon)/(\varepsilon^4 k)$. This implies that in Agrawal et al., the approximation is never better than $1 - 1/e - \tilde{O}(1/k^{1/5})$.*

# B  Missing proofs from Section 4

**Proposition 4.1.** *Fix subsets $G, B$ of elements (denoting "good" and "bad") such that $|G| = k$ and $|B| = n - k$; let $r \in [0, k]$ be some parameter. Let $m$ denote the size of the memory buffer, and let $p$ denote the probability that a random subset of size $m$ contains at least $r - 1$ good elements. Let $f : G \cup B \to \mathbb{R}$ be a function that satisfies the following symmetries:*

- *$f$ is symmetric over good (resp. bad) elements, namely there exists $\hat{f}$ such that*

$$f(S) = \hat{f}(|S \cap G|, |S \cap B|).$$

- *For any set $S$ with $\leq r - 1$ good elements, $f$ does not distinguish between good and bad elements, namely for $g \leq r - 1$,*

$$\hat{f}(g, b) = \hat{f}(0, b + g).$$

*Then any algorithm has expected value at most*

$$ALG \leq (1 - pk)\hat{f}(0, k) + pk \cdot OPT. \tag{7}$$

**Lemma 4.3** (exponential-universe coverage function (**new construction**)). *There exists a (monotone submodular) coverage function $f$ over an exponential universe $U$ that satisfies the desiderata of Proposition 4.1 for $r = 3$, and such that:*

- *$\hat{f}(0, k) = (1 - 1/e + o(1))|U|$.*
- *$OPT = f(G) = |U|$.*

*Proof.* The universe of elements to be covered $U$ is the $n$-dimensional $k$-side-length hypercube $[k]^n$. Let $B_i$ denote the set $\{\mathbf{x} \in [k]^n \mid x_i = 1\}$, namely, the set of all the vectors of which the $i$-th coordinate is 1. Let $G_i$ denote the set $\{\mathbf{x} \in [k]^n \mid x_1 = i, x_2 \neq k\} \cup \{\mathbf{x} \in [k]^n \mid x_1 = k, x_2 = k\}$. Our set system consists of $k$ good sets $G_1, \ldots, G_k$ and $n - k$ bad sets $B_{k+1}, \ldots, B_n$. We make the following three observations:

- For any $b \in [n-k]$ and $g \in \{0, 1, 2\}$, any distinct $i_1, \ldots, i_b \in \{k+1, \ldots, n\}$ and $j_1, j_2 \in [k]$, we have that

$$|(\cup_{t=1}^{b} B_{i_t}) \cup (\cup_{t=1}^{g} G_{j_t})| = \left(1 - \left(1 - \frac{1}{k}\right)^{b+g}\right) k^n.$$

- Moreover, it holds that

$$|\cup_{j=1}^{k} G_j| = \left(1 - \frac{1}{k} + \frac{1}{k^2}\right) k^n.$$

- Finally, the output of the coverage function is fully determined by the number of good sets and the number of bad sets in the input. Hence, there is a succinct encoding of all the possible values of this coverage function, which uses $O(\log n)$ bits. $\square$

**Theorem 4.4.** *Any $(1 - 1/e + \varepsilon)$-approximation algorithm in the random order strong oracle model must use the following memory:*

- *$\Omega(n)$ for a general monotone submodular function.*

- *$\Omega(n/k^2)$ for a coverage function over a polynomial universe.*

- *$\Omega(n/k^{3/2})$ for a coverage function over an exponential universe.*

*Proof.* Each case follows by combining Proposition 4.1 with Lemmata 4.1-4.3 respectively. For each case, we need to compute a bound on $m$ such that the probability $p$ of observing $r-1$ good elements in a random sample of $m$ is $p \leq \varepsilon/k$.

**Case $r = 2\varepsilon k$:** For $m = \varepsilon n$, the expected number of good elements is $km/n = \varepsilon k$. By Chernoff bound, probability of deviating by $\approx \varepsilon k$ is exponentially small.

**Case $r = 2$:** For $m = \varepsilon n/k^2$, the expected number of good elements is $km/n = \varepsilon/k$. By Markov's inequality, the probability of having at least $r - 1 = 1$ good element in memory is $p < \varepsilon/k$.

**Case $r = 3$:** For $m = \sqrt{\varepsilon} n/k^{3/2}$, each good element appears in memory with probability $\sqrt{\varepsilon} k^{-3/2}$. The probability that any fixed pair of good elements appear in memory is $\leq \varepsilon k^{-3}$. Taking a union bound over $\binom{k}{2}$ pairs, we have that $p < \varepsilon/k$. $\square$

## C A $(1/e - \varepsilon)$-approximation for non-monotone submodular maximization

In this section, we show that the basic algorithm described in Algorithm 2 can be altered to give a $1/e$-approximation to the cardinality constrained non-monotone case (Algorithm 4).

Algorithm 4 uses the same kind of multi-level scheme as Algorithm 2 but differs in two ways.

First, Algorithm 4 further sub-samples the elements of the input so that the probability of including any element is *exactly* $1/(\alpha k)$ lines 8–13 (coloured in orange). The sub-sampling allows us to bound the maximum probability that an element of the input is included in the solution. In particular, the sub-sampling is done by having the algorithm compute (on the fly) the conditional probability that an element $e$ could have been selected had it appeared in the past. This gives us the ability to compute an appropriate sub-sampling probability to ensure that $e$ does not appear in $H$ with too high a probability. In terms of the proof, the sub-sampling allows us to perform a similar analysis to the RANDOMGREEDY algorithm of Buchbinder et al. [BFNS14].[6]

Second, the addition of elements to levels in $[z_l, z_h]$ may cause a decrease in the function value, meaning that we no longer maintain the nesting property in Lemma A.1 (for similar reasons, line 22 also differs from the monotone case by simply copying level $\ell$ into $\ell + 1$). Fortunately, we only require the nesting property to hold on levels outside of $[z_l, z_h]$. We show that this remains true for Algorithm 4.

---

[6]A difference here is that instead of analysing a random element of the top-$k$ marginals, we analyse the optimal set directly.

---

**Algorithm 4** NONMONOTONESTREAM$(f, E, k, \alpha)$

1: Partition $E$ into windows $w_i$ for $i = 1, \ldots, \alpha k$ with Algorithm 1.
2: $L_\ell^0 \leftarrow \emptyset$ for $i = 0, \ldots, k$
3: $H \leftarrow \emptyset$
4: $x_e^i \leftarrow \text{Unif}(0, 1)$ for $e \in E, i \in [\alpha k]$
5: **for** $i = 1, \ldots, \alpha k$ **do**
6:     $C_i \leftarrow \emptyset$
7:     $z_l^i, z_h^i \leftarrow \max\{0, \lfloor i/\alpha \rfloor - 20\alpha\sqrt{k \log k}\}, \min\{k, \lceil i/\alpha \rceil + 20\alpha\sqrt{k \log k}\}$
8:     **for** $e \in$ window $w_i$ **do**
9:         **for** $j = 1, \ldots, i$ **do**
10:             Reconstruct $L_\ell^{j-1}$ for all $\ell$ from $\mathcal{H}_{i-1}$
11:             $A_e^j = \{r \mid 1 \leq r < j, \sum_{\ell=z_l^r}^{z_h^r} f(e|L_\ell^{r-1}) < f_r \text{ or } x_e^r > q_e^r\}$ (see line 16 for $f_r$)
12:             $q_e^j = \frac{\alpha k - j + |A_e^j| + 1}{\alpha k}$
13:         **if** $x_e^i \leq q_e^i$ **then** $C_i \leftarrow C_i \cup \{e\}$
14:     Sample each $e \in H$ with probability $\frac{1}{\alpha k}$ and add to $C_i$
15:     $e^\star \leftarrow \text{argmax}_{e \in C_i} \sum_{\ell=z_l^i}^{z_h^i} f(e|L_\ell^{i-1})$
16:     $f_i \leftarrow \sum_{\ell=z_l^i}^{z_h^i} f(e^\star|L_\ell^{i-1})$
17:     **if** $\sum_{\ell=z_l}^{z_h} f(L_\ell^{i-1} \cup \{e^\star\}) > \sum_{\ell=z_l}^{z_h} f(L_{\ell+1}^{i-1})$ **then**
18:         $H = H \cup \{e^\star\}$
19:         $L_{\ell+1}^i \leftarrow L_\ell^{i-1} \cup \{e^\star\}$ for all $\ell \in [z_l, z_h]$
20:         **for** $\ell = 1, 2 \ldots, k$ **do**
21:             **if** $f(L_\ell^i) \geq f(L_{\ell+1}^i)$ **then**
22:                 $L_{\ell+1}^i \leftarrow L_\ell^i$
23: **return** $\arg\max_\ell f(L_\ell^{\alpha k})$

---

**Lemma C.1.** *Let $z_l^i$ and $z_h^i$ be the value of $z_l$ and $z_h$ in Algorithm 4 on window $i$. For all $i$ and $\ell \notin [z_l^{i+1}, z_h^{i+1}]$, $f(L_\ell^i) \leq f(L_{\ell+1}^{i+1})$.*

*Proof.* Consider a window $i+1$. Regardless of whether an element $e^\star$ was added to the solution or not, levels $\ell > z_h^{i+1}$ and $\ell < z_l^{i+1}$ are not changed. Thus $L_{\ell+1}^{i+1} = L_{\ell+1}^i$, so $f(L_{\ell+1}^{i+1}) = f(L_{\ell+1}^i) \geq f(L_\ell^i)$ by line 21. Thus $f(L_{\ell+1}^{i+1}) \geq f(L_\ell^i)$ for $\ell \notin [z_l^{i+1}, z_h^{i+1}]$. $\square$

**Implementation of Algorithm 4** For clarity of exposition, we compute $x_e^i$ up front in line 4. However, we can compute them on the fly in practice since each element only uses its value of $x_e^i$ once (lines 11 and 13). This avoids an $O(n\alpha k)$ memory cost associated with storing each $x_e^i$. Finally, we assume that there are no ties when computing the best candidate element in each window (line 15). Ties can be handled by any arbitrary but consistent tie-breaking procedure. Any additional information used to break the ties (for example an ordering on the elements $e$) must be stored alongside $f_i$ for the computation of $A_e^j$ (line 11).

Next, we show that the probability an element $e$ is in a candidate set $C_i$ is exactly $1/(\alpha k)$ for any $e \in E$. The proof is conceptually very similar to Lemma C.2, in which we showed that $p_e^i := \Pr(e \in w_i \mid \mathcal{H}_{i-1}) \geq 1/(\alpha k)$. However, we make the additional observation that the proof of Lemma C.2 also offers a way for the algorithm to compute $\Pr(e \in w_i \mid \mathcal{H}_{i-1})$ exactly. By computing this probability and sub-sampling $e$ with probability $1/(\alpha k p_e^i)$, we ensure that $e$ is included in $C_i$ (line 13) with probability exactly $1/(\alpha k)$.

**Lemma C.2.** *Fix a history $\mathcal{H}_{i-1}$. For any element $e \in E$, we have $\Pr(e \in C_i \mid \mathcal{H}_{i-1}) = 1/(\alpha k)$.*

*Proof.* When $e \in \mathcal{H}_{i-1}$, $e$ is added to $C_i$ with probability $1/(\alpha k)$ exactly. Thus we assume $e \in E \setminus \mathcal{H}^{i-1}$.

Let $\mathcal{T} = A_e^i \cup \{i, i+1, \ldots, \alpha k\}$ (where $A_e^i$ is defined on line 11). Fix any $\mathcal{H}_{i-1}$-compatible partition $P$ and $j \in \mathcal{T}$. As in Lemma C.2, we begin by showing that we can create another $\mathcal{H}_{i-1}$-compatible

partition $\tilde{P}$ by setting $\tilde{P}(e) = j$ and all other values of $\tilde{P}$ equal to $P$. Since $\tilde{P}$ is equal to $P$ everywhere except on $e$, this maps each such partition $P$ to a unique partition $\tilde{P}$. Consequently, the map from $P$ to $\tilde{P}$ is a bijection, and so the number of $\mathcal{H}_{i-1}$-compatible partitions with $P(e) = j$ is equal for any $j \in \mathcal{T}$.

Observe that because $e \notin \mathcal{H}_{i-1}$, $e$ can be removed from window $P(e)$ without changing $\mathcal{H}_{i-1}$. We now separate the argument into two cases, for $j \geq i$ and $j \in A_e^i$. If $j \geq i$, the mapping of $P$ to $\tilde{P}$ does not change $\mathcal{H}_{i-1}$, since window $j$ is not included in the computations determining $\mathcal{H}_{i-1}$. Thus $\tilde{P}$ is trivially $\mathcal{H}_{i-1}$-compatible. If $j \in A_e^i$, this means that either $\sum_{\ell=z_l^j}^{z_h^j} f(e|L_\ell^{j-1})$ was too small, or it was probabilistically ignored because $x_e^j$ was too big. Either way, this means that $e \notin C_j$ and hence $e \notin L_\ell^j$ for any $\ell$. Consequently, adding $e$ to window $j$ does not change $\mathcal{H}_{i-1}$. As a result, $\tilde{P}$ is $\mathcal{H}_{i-1}$-compatible for any $j \in \mathcal{T}$.

For any windows $j, j' \in \mathcal{T}$, we then have

$$
\begin{aligned}
\Pr\left(e \in w_j | \mathcal{H}_{i-1}, e \notin \mathcal{H}_{i-1}\right) &= \frac{\#\text{partitions } P \text{ with } P(e) = j \text{ and } P \text{ is } \mathcal{H}_{i-1}\text{-compatible}}{\#\text{partitions } P \text{ is } \mathcal{H}_{i-1}\text{-compatible}} \\
&= \frac{\#\text{partitions } P \text{ with } P(e) = j' \text{ and } P \text{ is } \mathcal{H}_{i-1}\text{-compatible}}{\#\text{partitions } P \text{ is } \mathcal{H}_{i-1}\text{-compatible}} \\
&= \Pr\left(e \in w_{j'} | \mathcal{H}_{i-1}, e \notin \mathcal{H}_{i-1}\right).
\end{aligned}
$$

We now have the ingredients to compute $\Pr\left(e \in C_j | \mathcal{H}_{i-1}\right) = 1/(\alpha k)$. Any element $e \notin \mathcal{H}_{i-1}$ must appear in some window, so we have

$$
\begin{aligned}
1 &= \sum_{s=1}^{\alpha k} \Pr\left(e \in w_s | \mathcal{H}_{i-1}, e \notin \mathcal{H}_{i-1}\right) \\
&= \sum_{s \in \mathcal{T}} \Pr\left(e \in w_s | \mathcal{H}_{i-1}, e \notin \mathcal{H}_{i-1}\right) \\
&= |\mathcal{T}| \Pr\left(e \in w_i | \mathcal{H}_{i-1}, e \notin \mathcal{H}_{i-1}\right)
\end{aligned}
$$

Since $\Pr\left(e \in C_i | \mathcal{H}_{i-1}\right) = \Pr\left(e \in w_i | \mathcal{H}_{i-1}, e \notin \mathcal{H}_{i-1}\right) q_e^i$, we have $\Pr\left(e \in C_i | \mathcal{H}_{i-1}\right) = \frac{1}{|\mathcal{T}|} \cdot |\mathcal{T}|/(\alpha k) = 1/(\alpha k)$. $\qquad \square$

We are now ready to show the approximation guarantees of Algorithm 4. To do this, we borrow the following lemma from Buchbinder et al. [BFNS14]:

**Lemma C.3** (Lemma 2.2 [BFNS14]). *Let $f : 2^E \to \mathbb{R}_+$ be a submodular function. Further, let $R$ be a random subset of $T \subseteq E$ in which every element occurs with probability at most $p$ (not necessarily independently). Then, $\mathbf{E}f(R) \geq (1 - p)f(\emptyset)$.*

First, we note that the analysis of Lemma 2.4 up to the application of submodularity in Equation (4) still applies, leading to the following observation:

**Observation C.1.** *Let $\bar{\mathcal{L}}_i = \{z_l, z_l + 1, \ldots, z_h\}$ where $z_l$ and $z_h$ are defined in Algorithm 2. Conditioned on a history $\mathcal{H}_i$ and window $i + 1$ being active (the event $\mathcal{A}_{i+1}$),*

$$
\sum_{\ell \in \bar{\mathcal{L}}_{i+1}} \mathbf{E}[f(L_{\ell+1}^{i+1}) - f(L_\ell^i) \mid \mathcal{H}_i, \mathcal{A}_{i+1}] \geq \frac{1}{k} \sum_{\ell \in \bar{\mathcal{L}}_{i+1}} \mathbf{E}[f(O \cup L_\ell^i) - f(L_\ell^i) \mid \mathcal{H}_i].
$$

Now we relate the value of $f(O \cup L_\ell^i)$ to $f(O)$. As in Buchbinder et al. [BFNS14], this will involve showing that no element of the ground set is included into any of the levels $L_\ell^i$ with too high of a probability.

**Lemma C.4.** *For every $i$ and every $\ell$, the $\mathbf{E}[f(O \cup L_\ell^i)] \geq (1 - 1/(\alpha k))^i f(O)$ for all $\ell \leq k$.*

*Proof.* To be inserted into a partial solution $L_\ell^i$, an element $e$ must appear as part of the candidate set $C_i$ in Algorithm 4 or have appeared in $\mathcal{H}_{i-1}$. As shown in Lemma C.2, when conditioned on $e$ not

appearing in $\mathcal{H}_{i-1}$, $e$ appears in $C_i$ with probability exactly $1/(\alpha k)$. Thus $\Pr(e \in \mathcal{H}_i \mid e \notin \mathcal{H}_{i-1}) \leq 1/(\alpha k)$. By induction, we have

$$
\begin{aligned}
\Pr(e \notin \mathcal{H}_i) &= \Pr(e \notin \mathcal{H}_i \mid e \notin \mathcal{H}_{i-1}) \Pr(e \notin \mathcal{H}_{i-1}) \\
&\geq \left(1 - \frac{1}{\alpha k}\right) \Pr(e \notin \mathcal{H}_{i-1}) \\
&\geq \left(1 - \frac{1}{\alpha k}\right)^i
\end{aligned}
$$

where we assume by induction that $\Pr(e \notin \mathcal{H}_{i-1}) \geq \left(1 - \frac{1}{\alpha k}\right)^{i-1}$. After $i$ windows, any particular element of the input is in $\mathcal{H}_i$ (and $L_\ell^i$ for any $\ell$) with probability at most $1 - (1 - 1/(\alpha k))^i$.

Define the submodular function $g(S) = f(S \cup O)$. By Lemma C.3 and the reasoning above,

$$
\mathbf{E}[g(L_\ell^i)] \geq (1 - 1/(\alpha k))^i f(O). \qquad \square
$$

**Theorem 3.1.** *Algorithm 4 obtains a $(1/e - \varepsilon)$-approximation for maximizing a non-monotone function $f$ with respect to a cardinality constraint.*

*Proof.* Let $Z_i = \sum_{j=1}^i \mathbb{1}_{\mathcal{A}_i}$ be a random variable measuring the number of active windows up to window $i$. We follow an analysis similar to Theorem 2.5. For $\sigma := 20\sqrt{k \log k}$, Lemma 2.3 shows that w.h.p. $(1 - \mathrm{poly}(1/k))$,

$$
\forall i \quad Z_i \in [\bar{Z}_i \pm \sigma]. \tag{8}
$$

By the definition of $O$, $f(L_\ell^i) \leq f(O)$ for any $\ell$ and $i$ since $|L_\ell^i| \leq k$. Again, since (8) fails only with probability $\mathrm{poly}(1/k)$, we can condition on it while only having negligible effect of $\pm \mathrm{poly}(1/k)f(O)$ on $\mathbf{E}[f(L_\ell^i)]$ (for any $i, \ell$). For the rest of the proof, the inequalities will hold up to a $\pm \mathrm{poly}(1/k)f(O)$ term (which has no effect on the final asymptotic error guarantee).

Let $\bar{\mathcal{L}}_i := [\bar{Z}_i \pm \alpha\sigma]$ and $\mathcal{L}_i := [Z_i \pm (\alpha\sigma + \sigma)]$. To keep both non-negative, we denote $i' := i - \alpha(\alpha\sigma + \sigma) - \Theta(1)$ and consider the contribution from $i$ s.t. $i' \geq 0$.

In the non-monotone case, our goal will be to argue that:

$$
\frac{1}{|\mathcal{L}_i|} \sum_{\ell \in \mathcal{L}_i} \mathbf{E}[f(L_\ell^i) \mid \mathcal{A}_{i+1}] \geq \frac{i+1}{\alpha k}\left(1 - \frac{1}{\alpha k}\right)^i f(O) - O\left(\frac{i+1}{\alpha^2 k}\right) f(O). \tag{9}
$$

In particular, for $\tau := \alpha(k - \alpha\sigma - 2\sigma)$, we have that

$$
\frac{1}{|\mathcal{L}_i|} \sum_{\ell \in \mathcal{L}_\tau} \mathbf{E}[f(L_\ell^\tau)] \geq \left(1/e - O\left(\frac{1}{\alpha} + \frac{\alpha\sigma}{k}\right)\right) f(O).
$$

Since the solutions $L_\ell^i$ increase in value as $\ell$ increases, we also have $L_k^\tau \geq \left(1/e - O\left(\frac{1}{\alpha} + \frac{\alpha\sigma}{k}\right)\right) f(O)$.

We now prove Equation (9). To avoid cluttering the notation, every expectation in the equation below is conditioned on $\mathcal{H}_{i-1}$. The reasoning below follows along the same lines as the monotone case. Roughly speaking, $\mathcal{L}_i$ and $\bar{\mathcal{L}}_i$ largely overlap, so their averages are close up to $1/poly(k)$ factors. However, since $\bar{\mathcal{L}}_i$ is deterministically defined, this allows us to apply Observation C.1. For $i$

s.t. $i' \geq 0$, we have:

$$\frac{1}{|\mathcal{L}_{i+1}|} \sum_{\ell \in \mathcal{L}_{i+1}} \mathbf{E}[f(L_\ell^{i+1}) \mid \mathcal{A}_{i+1}] = \frac{1}{|\mathcal{L}_i|} \sum_{\ell \in \mathcal{L}_i} \mathbf{E}[f(L_{\ell+1}^{i+1}) \mid \mathcal{A}_{i+1}] \qquad \text{(Def. of } \mathcal{L}_i\text{)}$$

$$= \frac{1}{|\mathcal{L}_i|} \left( \sum_{\ell \in \mathcal{L}_i \setminus \bar{\mathcal{L}}_{i+1}} \mathbf{E}[f(L_{\ell+1}^{i+1}) \mid \mathcal{A}_{i+1}] + \sum_{\ell' \in \bar{\mathcal{L}}_{i+1}} \mathbf{E}[f(L_{\ell'+1}^{i+1}) \mid \mathcal{A}_{i+1}] \right) \qquad \text{(Eq. (8))}$$

$$= \frac{1}{|\mathcal{L}_i|} \left( \sum_{\ell \in \mathcal{L}_i \setminus \bar{\mathcal{L}}_{i+1}} \mathbf{E}[f(L_\ell^i) \mid \mathcal{A}_{i+1}] + \sum_{\ell' \in \bar{\mathcal{L}}_{i+1}} \mathbf{E}[f(L_{\ell'+1}^{i+1}) \mid \mathcal{A}_{i+1}] \right) \qquad \text{(Lemma C.1)}$$

$$\geq \frac{1}{|\mathcal{L}_i|} \left( \sum_{\ell \in \mathcal{L}_i} \mathbf{E}[f(L_\ell^i) \mid \mathcal{A}_{i+1}] + \sum_{\ell' \in \bar{\mathcal{L}}_{i+1}} \frac{1}{k} \mathbf{E}[(f(O \cup L_{\ell'}^i) - f(L_{\ell'}^i)) \mid \mathcal{A}_{i+1}] \right) \qquad \text{(Observation C.1)}$$

$$= \frac{1}{|\mathcal{L}_i|} \left( \sum_{\ell \in \mathcal{L}_i} \mathbf{E}[f(L_\ell^i)] + \sum_{\ell' \in \bar{\mathcal{L}}_{i+1}} \frac{1}{k} \mathbf{E}[(f(O \cup L_{\ell'}^i) - f(L_{\ell'}^i))] \right) \qquad (\mathcal{A}_{i+1} \text{ indep. of } L_\ell^i \ \forall \ell)$$

$$\geq \frac{1}{|\mathcal{L}_i|} \left( \sum_{\ell \in \mathcal{L}_i} \mathbf{E}[f(L_\ell^i)] + \sum_{\ell' \in \bar{\mathcal{L}}_{i+1}} \frac{1}{k} \mathbf{E}[(1 - (\alpha k)^{-1})^i f(O) - f(L_{\ell'}^i)] \right) \qquad \text{(Lemma C.4)}$$

$$\geq \frac{1}{|\mathcal{L}_i|} \sum_{\ell \in \mathcal{L}_i} \left( \frac{(1 - (\alpha k)^{-1})^i}{k} f(O) + \frac{k-1}{k} \mathbf{E}[f(L_\ell^i)] \right) - O\left( \frac{\sigma}{k|\mathcal{L}_i|} \right) f(O) \qquad (|\mathcal{L}_i| - |\bar{\mathcal{L}}_{i+1}| = 2\sigma)$$

$$= \frac{(1 - (\alpha k)^{-1})^i}{k} f(O) + \left( 1 - \frac{1}{k} \right) \frac{1}{|\mathcal{L}_i|} \sum_{\ell \in \mathcal{L}_i} \mathbf{E}[f(L_\ell^i)] - O\left( \frac{\sigma}{k|\mathcal{L}_i|} \right) f(O).$$

The main difference between the non-monotone and monotone case is the application of Lemma C.4 (to bound the maximum amount an element may hurt the solution).

To address the conditioning on $\mathcal{A}_{i+1}$, consider

$$\frac{1}{|\mathcal{L}_{i+1}|} \sum_{\ell \in \mathcal{L}_{i+1}} \mathbf{E}[f(L_\ell^{i+1})] = \frac{1 - \Pr[\mathcal{A}_{i+1}]}{|\mathcal{L}_{i+1}|} \sum_{\ell \in \mathcal{L}_{i+1}} \mathbf{E}[f(L_\ell^{i+1}) \mid \neg\mathcal{A}_{i+1}] + \frac{\Pr[\mathcal{A}_{i+1}]}{|\mathcal{L}_{i+1}|} \sum_{\ell \in \mathcal{L}_{i+1}} \mathbf{E}[f(L_\ell^{i+1}) \mid \mathcal{A}_{i+1}]$$

$$\geq \left( 1 - \frac{\Pr[\mathcal{A}_{i+1}]}{k} \right) \frac{1}{|\mathcal{L}_i|} \sum_{\ell \in \mathcal{L}_i} \mathbf{E}[f(L_\ell^i)] + \frac{(1 - (\alpha k)^{-1})^i}{k} \Pr[\mathcal{A}_{i+1}] f(O) - O\left( \frac{\sigma}{k|\mathcal{L}_i|} \right) \Pr[\mathcal{A}_{i+1}] f(O)$$

$$\geq \left( 1 - \frac{1}{\alpha k} \right) \frac{1}{|\mathcal{L}_i|} \sum_{\ell \in \mathcal{L}_i} \mathbf{E}[f(L_\ell^i)] + \frac{(1 - (\alpha k)^{-1})^i}{\alpha k} f(O) - O\left( \frac{1}{\alpha^2 k} \right) f(O).$$

The second line requires $\mathbf{E}[f(L_\ell^{i+1}) \mid \neg\mathcal{A}_{i+1}] \geq \mathbf{E}[f(L_\ell^i) \mid \neg\mathcal{A}_{i+1}]$. Fortunately, this is true as Algorithm 4 only increases the average of values in $\mathcal{L}_i$: levels $\bar{\mathcal{L}}_i \subseteq \mathcal{L}_i$ are only updated if an increase is detected on line 17. The last line follows from plugging in $|\mathcal{L}_i| = 2(\alpha + 1)\sigma$ and $1 - e^{-1/\alpha} \leq \Pr[\mathcal{A}_{i+1}] = 1 - (1 - \frac{1}{\alpha k})^k \leq \frac{1}{\alpha}$. Now we may remove the (implicit) conditioning on $\mathcal{H}_{i-1}$, as everything has been conditioned on it so far.

Let $\bar{f}_i = \frac{1}{|\mathcal{L}_i|} \sum_{\ell \in \mathcal{L}_i} \mathbf{E}[f(L_\ell^i)]$. Now we show by induction that $\bar{f}_i \geq \frac{i}{\alpha k} \left( 1 - \frac{1}{\alpha k} \right)^{i-1} f(O) - O\left( \frac{i}{\alpha^2 k} \right) f(O)$. The base case is clearly true, as the first window has probability $1/(\alpha k)$ of catching any optimal element. By our analysis from above:

$$\bar{f}_{i+1} \geq \left( 1 - \frac{1}{\alpha k} \right) \bar{f}_i + \frac{(1 - (\alpha k)^{-1})^i}{\alpha k} f(O) - O\left( \frac{1}{\alpha^2 k} \right) f(O)$$

$$\geq \left( 1 - \frac{1}{\alpha k} \right) \left( \frac{i}{\alpha k} \left( 1 - \frac{1}{\alpha k} \right)^{i-1} - O\left( \frac{i}{\alpha^2 k} \right) \right) f(O) + \frac{(1 - \frac{1}{\alpha k})^i}{\alpha k} f(O) - O\left( \frac{1}{\alpha^2 k} \right) f(O)$$

$$\geq \frac{i+1}{\alpha k} \left( 1 - \frac{1}{\alpha k} \right)^i f(O) - O\left( \frac{i+1}{\alpha^2 k} \right) f(O).$$

In particular, for $\tau := \alpha(k - \alpha\sigma - 2\sigma)$, we have that

$$\frac{1}{|\mathcal{L}_i|} \sum_{\ell \in \mathcal{L}_\tau} \mathbf{E}[f(L_\ell^\tau)] \geq \left(1/e - O\left(\frac{1}{\alpha} + \frac{\alpha\sigma}{k}\right)\right) f(O).$$

Setting $\alpha = \Theta(1/\varepsilon)$ where $\varepsilon = \omega\left(\frac{\log k}{k}\right)^{1/4}$ gives the desired result. $\qquad\square$

We remark that Algorithm 4 also achieves a guarantee of $1 - 1/e - \varepsilon$ for the monotone case, as Lemma 2.4 and Theorem 2.5 both still apply to Algorithm 4 when $f$ is monotone. The main difference between the two is the sub-sampling procedure (lines 8–13), which increases the running time of the algorithm.

## D   Experiments

All code can be found at `https://github.com/where-is-paul/submodular-streaming` and all datasets can be found at `https://tinyurl.com/neurips-21`.

**Experiments for non-monotone submodular streaming**

For non-monotone submodular functions, we compare against the offline random greedy algorithm of Buchbinder et al. [BFNS14].

**Datasets**   Our datasets are drawn from diversity maximization tasks described in [LSK$^+$21]. Here, given an $n \times n$ matrix $L$, the task is to find a subset $S$ of $k$ indices such that $\log\det(L_S)$ is maximized.[7] Since our non-monotone algorithm is significantly more expensive to run than our monotone algorithm, we created substreams from these datasets by sampling a consecutive run of 1024 stream elements at random and then permuting them. As in the monotone case, for each data set we run the standard offline algorithm (random greedy) and compare against our streaming algorithm with $k$ varying from 1 to 10. Table 3 describes the data sources. Figure 2 shows the performance of the three algorithms on each data set.

| dataset | source |
|---------|--------|
| gowalla | gowalla geolocation data |
| yahoo!  | yahoo front page visit data |
| bing    | anonymized search data from the bing search engine |

Table 3: Description of data sources for non-monotone experiments.

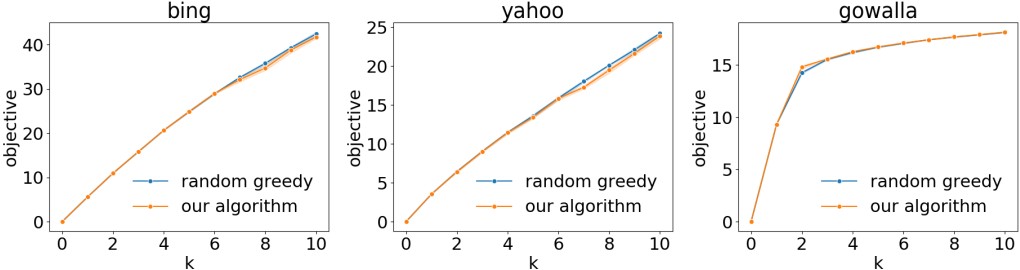

Figure 2: Performance of standard greedy, random greedy, and our algorithm on each data set (averaged across 10 runs, shaded regions represent variance across different random orderings).

---

[7]$L_S$ is the $k \times k$ submatrix formed by taking entries from $L$ with rows and columns in $S$.