# OpenReview forum: "Cardinality constrained submodular maximization for random streams"
_NeurIPS.cc/2021/Conference — NeurIPS 2021 Poster_

### Official Review · Reviewer_GbQb · 2021-07-12

**Rating:** 7
**Confidence:** 3

**Summary:**

This paper studies the problem of maximizing a submodular function. For monotone submodular functions subject to a cardinality constraint, the paper gives a $(1-1/e-\epsilon)$-approximation. The space requirement is $O(k/\epsilon)$ where $k$ is the cardinality constraint. This is in contrast to a previous algorithm of Agrawal et al. which gave an algorithm with the same approximation ratio but with an exponential dependence on $\epsilon$. Hence, the new algorithm obtains an exponential improvement on the dependence on $\epsilon$. The new algorithm is also straightforward to implement. For the case of non-monotone submodular maximization subject to a cardinality constraint of size $k$, the authors gives an algorithm that achieves a $(1/e - \epsilon)$-approximation with space complexity $O(k / \epsilon)$.

The authors also prove a $(1-1/e+\epsilon)$ hardness result for monotone submodular maximization. This improves upon the previous best hardness result of $0.7$.

Finally, the authors show some experimental results and it is apparent that their algorithm performs quite well when compared to offline greedy.

**Limitations And Societal Impact:**

Yes

**Main Review:**

**Strengths**
- The problem is interesting and well-motivated.
- In the case of monotone submodular maximization, the present paper significantly improves over previously known results in terms of space requirements. This makes the algorithm a practical one.
- In the case of non-monotone submodular maximization, the paper also gives a significant improvement over known results - in this setting, the approximation ratio is improved.
- The algorithmic techniques are quite nice. Although it is straightforward to implement, it is quite non-trivial.
- The hardness is improved from $0.7$ to a tight $1-1/e$.
- The authors run some experiements and their algorithm shows good empirical performance.
- The paper is written well.

**Weaknesses** \
No complaints about the paper.

**Target Audience** \
The paper will be of interest to other researchers working in submodular optimization.


**Time Spent Reviewing:**

2

---

> ### Author Response · Authors · 2021-08-09
> **We thank the reviewer for their review!**
>
> Thank you for your feedback!

---

> > ### Comment · Reviewer_GbQb · 2021-08-31
> > **Thanks**
> >
> > Just acknowledging I've read the response. :)
> >
> > Review remains unchanged, of course.

---

### Official Review · Reviewer_QQMD · 2021-07-15

**Rating:** 8
**Confidence:** 4

**Summary:**

This work studies cardinality-constrained submodular maximization in a one-pass
streaming model (with a random ordering). The main results are:
- A $(1-1/e-\varepsilon)$-approximation algorithm for monotone functions that
  uses $O(k/\varepsilon)$ memory, where $k$ is the cardinality constraint.
- A $1/e$-approximation algorithm for non-monotone submodular functions that
  also uses $O(k / \varepsilon)$ memory.
- A new hardness result which shows that any
  $O(1-1/e+\varepsilon)$-approximation algorithm requires $\Omega(n)$ memory.

The algorithm for monotone functions in this paper improves on the recent work
of [Agrawal-Shadravan-Stein, ITCS 2019] by removing an exponential dependency
on $\varepsilon$ in the space complexity. Similarly, they close the previously
best-known streaming hardness gap of $7/8$ to $(1-1/e+\varepsilon)$, which was
also proven in [Agrawal-Shadravan-Stein, ITCS 2019]. Lastly, the authors provide
a strong set of experiments that demonstrates their streaming algorithm is very
competitive with an offline greedy algorithm.

**Main Review:**

**Originality.**
This work directly builds on the recent work of [Agrawal-Shadravan-Stein, ITCS
2019], and combines several techniques from the submodular maximization
literature to achieve state-of-the-art results when the elements arrive in a
random stream. The originality of this paper is mostly due to the results the
authors achieved, as opposed to a new technique or hammer for future use.

**Quality.**
This work is very high quality. It would be helpful to give the reader more
context around the results in Section 4, but aside from that this is a nice,
self-contained paper.

**Clarity.**
This paper is very well written and easy to follow.

**Significance.**
This paper is a great addition to the streaming submodular maximization
literature. Several recent works [NTM+18 (ICML), KMZ+19 (ICML), FNSZ20 (STOC)]
have studied submodular maximization in an adversarial stream, and [ASS19
(ITCS)] recently explored what happens when the stream is a random permutation
of the elements. This work answers many of the open questions in [ASS19].

**Typos / Suggestions.**
- [page 4] Typo: "a evolving" --> "an evolving"
- [page 4] Typo: "collection of partial solution" --> "collection of partial solutions"
- [page 5] $f(O)$ is used before $O$ is introduced as the OPT solution.

**Time Spent Reviewing:**

3

---

> ### Author Response · Authors · 2021-08-09
> **We thank the reviewer for their review!**
>
> We will correct the typos in the final version.

---

### Official Review · Reviewer_zv6j · 2021-07-18

**Rating:** 7
**Confidence:** 3

**Summary:**

This paper considers cardinality constrained submodular maximization, both monotone and non-monotone, in the random streaming setting. In the random streaming setting the universe is received in a uniformly random permutation, as opposed to the adversarial setting where the universe could be received in any order.

In the random streaming setting, better approximation guarantees are possible compared to the adversarial setting (where 1/2 is the best possible). An algorithm developed by [ASS19] was shown to give a $1-1/e-\epsilon$ guarantee for cardinality constrained monotone submodular maximization, but has memory exponential in $\epsilon$ making it impractical. This paper introduces a streaming algorithm that gives the $1-1/e-\epsilon$ needing only $O(k/\epsilon)$ memory. They also prove $1-1/e$ is the best possible. Further, they introduce a streaming algorithm for the non-monotone version of the problem that gives a $1/e$ approximation guarantee also in $O(k/\epsilon)$ memory.

The algorithms are analyzed empirically, and shown to be practical.

**Limitations And Societal Impact:**

Yes

**Main Review:**

I think this is a good paper. There are clear theoretical improvements compared to related work. In particular, the memory is much improved compared to [ASS19] while maintaining equally good approximation ratio. In addition, it looks like this is the first paper to consider non-monotone SM in the random stream model. The techniques used to get these results look interesting to me.

Originality: [ASS19] is the only work that I know of that is very closely related, and this appears to have sufficiently original ideas compared to [ASS19], and has made a substantial improvement on the memory. The paper appears very original compared to works other than [ASS19] that I know of.

Quality: The paper appears to be of good quality.

Clarity: This paper is exceptionally well-written.

Significance: The results are definitely of interest to the submodular optimization community. One limitation to its significance is that the approximation guarantees assume the random stream model, which may be relatively limiting, but I think the results are still significant.

Weaknesses: The experimental results are only showing the objective value of the different algorithms. Why isn't the algorithm compared to SieveStreaming on memory used (or maybe it should be SieveStreaming++ if that is more efficient as far as memory)? Also, why not compare to [NTM+18]? In addition, in what sort of applications would we have a random streaming model?

**Time Spent Reviewing:**

2

---

> ### Author Response · Authors · 2021-08-09
> **We will include additional experiments in the final version**
>
> We thank the reviewer for their review!
>
> As per the reviewer's suggestion, we plan to provide additional comparisons to [NTM+18] in the final version. As noted, we currently do not perform experiments on SieveStreaming for memory used. One reason is that the memory use of SieveStreaming is asymptotically higher ($O(k \log k / \varepsilon)$ vs $O(k / \varepsilon)$). We can further expand on the difference between the two in the final version.
>
> Finally, we also plan to expand on the applications of the random order streaming model. One important application is that the random order model generalizes models where the input is drawn i.i.d. from a distribution (the iid assumption is used in almost all ML settings). In particular, the model allows the input to be drawn from an *unknown* distribution. That is, our algorithm works out of the box without seeing any samples of the distribution beforehand. In addition, we note that worse case order may be overly pessimistic for most applications -- elements in practice are typically somewhat closer to the random order case. An examination of our algorithm for semi-adversarial streams would be an interesting topic for future work.

---

### Official Review · Reviewer_EsiB · 2021-07-20

**Rating:** 7
**Confidence:** 4

**Summary:**

The paper studies the submodular maximization problem with a cardinality constraint in the random stream model where the algorithm makes a single pass over a random permutation of the elements. The main contribution of the paper is an algorithm for monotone functions with a nearly-optimal 1-1/e-eps approximation and O(k/eps) memory. The algorithm also extends to non-monotone functions via sub-sampling, giving a 1/e-eps approximation  using the same memory requirement. On the hardness side, the paper shows an unconditional hardness result showing that an approximation of 1+1/e is best possible for any single-pass random stream algorithm, even if the algorithm is allowed unbounded computation time.

**Limitations And Societal Impact:**

Yes

**Main Review:**

Significance: The paper studies a well-motivated and widely studied problem and it achieves significant improvements over the state of the art. The proposed algorithms build on the prior work that shows an analogous approximation but using memory and time that is exponential in k and eps. In contrast, the proposed algorithm has essentially ideal memory requirement and the algorithm is efficient and practical. The experiments show that the algorithm outperforms the state of the art algorithm for arbitrary streams and its performance is close to that of the offline greedy algorithm. Overall, both the algorithmic and hardness contributions are a substantial addition to this line of work.

Novelty/originality: The algorithms build on prior work but there is substantial novelty in the design and analysis of the algorithm that leads to the significantly improved memory and time efficiency. The algorithm is simple and easy to implement, and the main complexity lies in the analysis, which is intricate and requires novel ideas.

Clarity: The paper is well-written and clear. The paper provides ample intuition for the algorithm and the analysis, which make the paper accessible despite the fact that the analysis is intricate.

**Time Spent Reviewing:**

I did not track the hours

---

> ### Author Response · Authors · 2021-08-09
> **We thank the reviewer for their review!**
>
> Thank you for the feedback!

---

### Decision · Program_Chairs · 2021-09-27

**Decision:**

Accept (Poster)

**Comment:**

The reviewers all agree that the paper makes substantial improvement over previous works in terms of memory requirement for an important problem.